# Benign Overfitting in Classification: Provably Counter Label Noise with Larger Models

**Kaiyue Wen**[1,*]**, Jiaye Teng**[1,2,3,*]**, Jingzhao Zhang**[1,2,3,†]
[1]Institute for Interdisciplinary Information Sciences, Tsinghua University
[2]Shanghai Qizhi Institute
[3]Shanghai Artificial Intelligence Laboratory
`{wenky20,tjy20}@mails.tsinghua.edu.cn, jingzhaoz@mail.tsinghua.edu.cn`

## Abstract

Studies on benign overfitting provide insights for the success of overparameterized deep learning models. In this work, we examine whether overfitting is truly benign in real-world classification tasks. We start with the observation that a ResNet model overfits benignly on Cifar10 but **not** benignly on ImageNet. To understand why benign overfitting fails in the ImageNet experiment, we theoretically analyze benign overfitting under a more restrictive setup where the number of parameters is not significantly larger than the number of data points. Under this mild overparameterization setup, our analysis identifies a phase change: unlike in the previous heavy overparameterization settings, benign overfitting can now fail in the presence of label noise. Our analysis explains our empirical observations, and is validated by a set of control experiments with ResNets. Our work highlights the importance of understanding implicit bias in underfitting regimes as a future direction.

## 1 Introduction

Modern deep learning models achieve good generalization performances even with more parameters than data points. This surprising phenomenon is referred to as benign overfitting, and differs from the canonical learning regime where good generalization requires limiting the model complexity (Mohri et al., 2018). One widely accepted explanation for benign overfitting is that optimization algorithms benefit from implicit bias and find good solutions among the interpolating ones under the overparametrized settings. The implicit bias can vary from problem to problem. Examples include the min-norm solution in regression settings or the max-margin solution in classification settings (Gunasekar et al., 2018a; Soudry et al., 2018; Gunasekar et al., 2018b). These types of bias in optimization can further result in good generalization performances (Bartlett et al., 2020; Zou et al., 2021; Frei et al., 2022). These studies provide novel insights, yet they sometimes differ from the deep learning practice: state-of-the-art models, despite being overparameterized, often do not interpolate the data points (*e.g.*, He et al. (2016); Devlin et al. (2018)).

We first examine the existence of benign overfitting in realistic setups. In the rest of this work, we term *benign overfitting* as the observation that *validation performance does not drop while the model fits more training data points*[1]. We test whether ResNet (He et al., 2016) models overfit data benignly for image classification on CIFAR10 and ImageNet. Our results are shown in Figure 1 below.

In particular, we first trained ResNet18 on CIFAR10 for 200 epochs and the model interpolates the training data. In addition, we also trained ResNet50 on ImageNet for 500 epochs, as opposed to the common schedule that stops at 90 epochs. Surprisingly, we found that although benign overfitting happens on the CIFAR10 dataset, *overfitting is not benign on the ImageNet dataset—the test loss increased as the model further fits the train set.* More precisely, the ImageNet experiment does not overfit benignly since the best model is achieved in the middle of the training. The different

---

[*]Equal contribution.
[†]Corresponding author.
[1]A more detailed discussion can be found in Appendix E.2. This definition is slightly different from existing theoretical literature but can be verified more easily in practice.

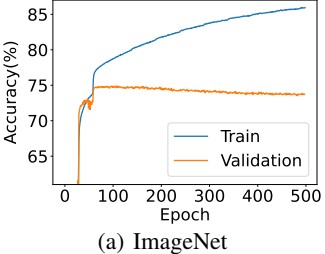 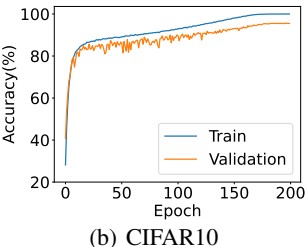

(a) ImageNet          (b) CIFAR10

Figure 1: **Different Overfitting Behaviors on ImageNet and CIFAR10.** We use ResNet50/ResNet18 to train ImageNet/CIFAR10 and plot the training loss as well as the validation loss. We find that ResNet50 overfits the ImageNet non-benignly, while ResNet18 overfits CIFAR10 benignly.

|  | Classification Noiseless | Classification Noisy | Regression Noisy |
|---|---|---|---|
| Mild over-parameterization $p = \Theta(n)$ | **Benign** (Ours) | **Not Benign** (Ours) | Not Benign (Bartlett et al., 2020) |
| Heavy over-parameterization $p = \omega(n)$ | Benign (Cao et al., 2021) (Wang et al., 2021a) | Benign (Chatterji et al., 2021) (Frei et al., 2022) (Wang et al., 2021a) | Benign (Bartlett et al., 2020) (Zou et al., 2021) |

Table 1: For classification problems, previous work (Cao et al., 2021; Chatterji et al., 2021) provided upper bounds under the heavy overparameterization setting. This paper focuses on a mild overparameterization setting, and shows that the label noise may break benign overfitting. Similar results were known in regression problems. For regression with noisy responses, Bartlett et al. (2020) shows that the interpolator fails under mild overparameterization regimes while may work under heavy overparameterization, as is consistent with the double descent curve. However, the analysis under mild overparameterization in the classification task remained unknown.

overfitting behaviors cannot be explained by known analysis for classification tasks, as no negative results have been studied yet.

Motivated by the above observation, our work aims to understand the cause for the two different overfitting behaviors in ImageNet and CIFAR10, and to reconcile the empirical phenomenon with previous analysis on benign overfitting. Our first hint comes from the level of overparameterization. Previous results on benign overfitting in the classification setting usually requires that $p = \omega(n)$, where $p$ denotes the number of parameters and $n$ denotes the training sample size (Wang et al., 2021a; Cao et al., 2021; Chatterji et al., 2021; Frei et al., 2022). However, in practice many deep learning models fall in the *mild overparameterization* regime, where the number of parameters is only slightly larger than the number of samples despite overparameterization. In our case, the sample size $n = 10^6$ in ImageNet, whereas the parameter size $p \approx 10^7$ in ResNets.

To close the gap, we study the overfitting behavior of classification models under *mild overparameterization* setups where $p = \Theta(n)$ (this is sometimes referred to as the asymptotic regimes). In particular, following Wang et al. (2021a); Cao et al. (2021); Chatterji et al. (2021); Frei et al. (2022), we analyze the solution of stochastic gradient descent for the Gaussian mixture models. We found that a **phase change** happens when we move from $p = \Omega(n \log n)$ (studied in Wang et al. (2021a)) to $p = \Theta(n)$. Unlike previous analysis, we show that benign overfitting now provably fails in the presence of label noise (see Table 1 and Figure 2). This aligns with our empirical findings as ImageNet is known to suffer from mislabelling and multi-labels (Yun et al., 2021; Shankar et al., 2020).

More specifically, our analysis (see Theorem 3.1 for details) under the *mild overparameterization (p = $\Theta(n)$)* setup supports the following statements that align with our empirical observations in Figure 1 and 2:

- When the labels are noiseless, benign overfitting holds under similar conditions as in previous analyses.
- When the labels are noisy, the interpolating solution can provably lead to a positive excess risk that does not diminish with the sample size.

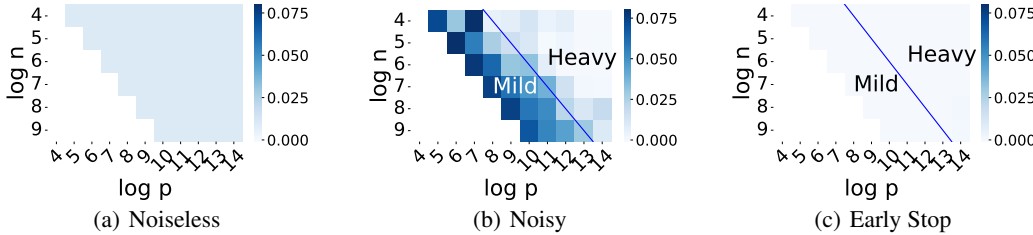

Figure 2: **Phase Transition in Noisy Regimes: parameter size $p$ vs sample size $n$.** We conduct experiments on simulated Gaussian mixture model (GMM) with signal strength $|\mu| = 40$ and noise level $\sigma = 1$. We use SGD to train a linear classifier and plot the excess risk. The brighter grid means that the classifier has smaller excess error, and therefore does not overfit. Figure (a) and Figure (b) correspond to the classifier at the last epoch. Figure (c) corresponds to the best classifier through the course of training.

- When the labels are noisy, early stopping can provably lead to better generalization performances compared to interpolating models.

Our analysis is supported by both synthetic and deep learning experiments. First, we train a linear classifier in Figure 2. Figure (a) shows that training on noiseless data does not cause overfitting, Figure (b) shows that training on noisy data with a label noise ratio 0.4 causes overfitting with mild overparameterization, and Figure (c) shows that applying early-stopping on the same noisy data can effectively avoid overfitting. Besides, we find that overfitting consistently happens under noisy regimes, despite the theoretical guarantee that the model overfits benignly when $p = \omega(n)$.

Furthermore, our analysis is validated by control experiments in Section 4 with ResNets. We first show that injecting random labels into the CIFAR10 dataset indeed leads to the failure of benign overfitting. We further show that, surprisingly, increasing the width of ResNet alone can alleviate overfitting incorrect labels, as our theory predicted.

More broadly, the empirical and theoretical results in our work point out that modern models may not operate under benign interpolating regimes. Hence, it highlights the importance of characterizing implicit bias when the model, though mildly overparameterized, does not interpolate the training data.

## 2 RELATED WORKS

Although the benign overfitting phenomenon has been systematically studied both empirically (Belkin et al., 2018b; 2019; Nakkiran et al., 2021b) and theoretically (see later), our work differs in that we aim to understand why benign overfitting fails in the classification setup. In particular, among all theoretical studies, the closest ones to us are Chatterji et al. (2021), Wang et al. (2021b) and Cao et al. (2021), as the analyses are done under the classification setup with Gaussian mixture models. Our work provides new results by moving from heavy overparameterization $p = \omega(n)$ to mild overparameterization $p = \Theta(n)$. We found that this new condition leads to an analysis that is consistent with our empirical observations, and can explain why benign overfitting fails. In short, compared to all previous studies on benign overfitting in classification tasks, we focus on *identifying the condition that breaks benign overfitting in our ImageNet experiments.* The comparison is summarized in Table 1.

**More related works on benign overfitting:** Researchers have made a lot of efforts to generalize the notation of benign overfitting beyond the novel work on linear regression (Bartlett et al., 2020), *e.g.*, variants of linear regression (Tsigler et al., 2020; Muthukumar et al., 2020; Zou et al., 2021; Xu et al., 2022), linear classification (Liang et al., 2020a; Belkin et al., 2018a) with different distribution assumptions (instead of Gaussian mixture model), kernel-based estimators (Liang et al., 2018; 2020b; Mei et al., 2022), neural networks (Frei et al., 2022). Among them, Cao et al. (2022) also identifies a phase change in benign overfitting when the overfitting can be provably harmful with a different data distribution under noiseless regimes.

**Gaussian Mixture Model** (GMM) represents the data distribution where the input is drawn from a Gaussian distribution with different centers for each class. The model was widely studied in hidden Markov models, anomaly detection, and many other fields (Hastie et al., 2001; Xuan et al., 2001; Reynolds, 2009; Zong et al., 2018). A closely related work is Jin (2009) which analyzes the lower bound for excess risk of the Gaussian Mixture Model under noiseless regimes. However, their analysis cannot be directly extended to either the overparameterization or noisy label regimes. Another closely related work (Mai et al., 2019) focuses on the GMM setting with mild overparameterization, but they require a noiseless label regime and rely on a small signal-to-noise ratio on the input, and thus cannot be generalized to our theoretical results.

**Asymptotic (Mildly Overparameterized) Regimes.** This paper considers asymptotic regimes where the ratio of parameter dimension and the sample size is upper and lower bounded by absolute constants. Previous work studied this setup with different focus than benign overfitting (*e.g.*, double descent), both in regression perspective (Hastie et al., 2019) and classification perspective (Sur et al., 2019; Mai et al., 2019; Deng et al., 2019). We study the mild overparameterization case because it generally happens in realistic machine learning tasks, where the number of parameters exceeds the number of training samples but not extremely.

## 3   OVERFITTING UNDER MILD OVERPARAMETERIZATION

We observed in Figure 1 that training ResNets on CIFAR10 and ImageNet can result in different overfitting behaviors. This discrepancy was not reflected in the previous analysis of benign overfitting in classification tasks. In this section, we provide a theoretical analysis by studying the Gaussian mixture model under the mild overparameterization setup. Our analysis shows that mild overparameterization along with label noise can break benign overfitting.

### 3.1   OVERPARAMETERIZED GAUSSIAN MIXTURE MODEL

In this subsection, we study the generalization performance of linear models on the Gaussian Mixture Model (GMM). We assume that linear models are obtained by solving logistic regression with stochastic gradient descent (SGD). This simplified model may help explain phenomenons in neural networks, since previous works show that neural networks converge to linear models as the width goes to infinity (Arora et al., 2019; Allen-Zhu et al., 2019).

We next introduce GMM under two setups of overparameterized linear classification: the noiseless regime and the noisy regime.

**Noiseless Regime.** Let $y \sim \text{Unif}\{-1, 1\} \in \mathbb{R}$ denote the ground truth label. The corresponding feature is generated by

$$\boldsymbol{x} = y\boldsymbol{\mu} + \boldsymbol{\epsilon} \in \mathbb{R}^p,$$

where $\boldsymbol{\epsilon}$ denotes noise drawn from a subGaussian distribution, and $\boldsymbol{\mu} \in \mathbb{R}^p$ denotes the signal vector. We denote the dataset $\mathcal{D} = \{(\boldsymbol{x}_i, y_i)\}_{i \in [n]}$ where $(\boldsymbol{x}_i, y_i)$ are generated by the above mechanism.

**Noisy Regime with contamination rate $\rho$.** For the noisy regime, we first generate noiseless data $(\boldsymbol{x}, y)$ using the noiseless regime, and then consider the data point $(\boldsymbol{x}, \tilde{y})$ where $\tilde{y}$ is the contaminated version of $y$. Formally, given the contamination rate $\rho$, the contaminated label $\tilde{y} = -y$ with probability $\rho$ and $\tilde{y} = y$ with probability $1 - \rho$. And the returned dataset is $\tilde{\mathcal{D}} = \{(\boldsymbol{x}_i, \tilde{y}_i)\}$.

For simplicity, we assume that the data points in the train set are linearly separable in both noiseless and noisy regimes. This assumption holds almost surely under mild overparameterization. Besides, we make the following assumptions about the data distribution:

**Assumption 1** (Assumptions on the data distribution)**.**

> *A1  The feature noise $\boldsymbol{\epsilon}$ is drawn from the Gaussian distribution, i.e., $\boldsymbol{\epsilon} \sim \mathcal{N}(0, \sigma^2 I)$.*
> *A2  The signal-to-noise ratio satisfies $\frac{\|\boldsymbol{\mu}\|}{\sigma} \geq c\sqrt{\log n}$ for a given constant $c$.*
> *A3  The ratio $p/n = r > 1$ is a fixed constant.*

The three assumptions are all crucial but can be made slightly more general (see Section 5). The first Assumption [A1] stems from the requirement to derive a lower bound for excess risk under a noisy regime. The second Assumption [A2] is widely used in the analysis (Chatterji et al., 2021; Frei et al.,

2022). For a smaller ratio, the model may be unable to learn even under the noiseless regime and return vacuous bounds. The third Assumption [A3] differs from the previous analysis, where we consider a mild overparameterization instead of heavy overparameterization (i.e., $p = \omega(n)$).

## 3.2 TRAINING PROCEDURE

We consider the multi-pass SGD training with logistic loss $\ell(\boldsymbol{w}; \boldsymbol{x}, y) = \log(1 + \exp(-y\boldsymbol{x}^\top \boldsymbol{w}))$. During each epoch, each data is visited exactly once randomly without replacement. Formally, at the beginning of each epoch $E$, we uniformly random sample a permutation $P_E : \{1, ..., n\} \to \{1, ..., n\}$, then at iteration $t$, given the learning rate $\eta$ we have

$$\boldsymbol{w}(t + 1) = \boldsymbol{w}(t) - \eta \nabla l(\boldsymbol{w}(t); \boldsymbol{x}(t), y(t)),$$

where $\boldsymbol{x}(t) = \boldsymbol{x}_{P_E(i)}, y(t) = y_{P_E(i)}$, given $t = nE + i, 1 \le i \le n$.

Under the above procedure, Proposition 3.1 shows that the classifier under the GMM regime with multi-pass SGD training will converge in the direction of the max-margin interpolating classifier.

**Proposition 3.1** (Interpolator of multi-pass SGD under GMM regime, from Nacson et al. (2019)). *Under the regime of GMM with logistic loss, denote the iterates in multi-pass SGD by $\boldsymbol{w}(t)$. Then for any initialization $w(0)$, the iterates $\boldsymbol{w}(t)$ converges to the max-margin solution almost surely, namely,*

$$\lim_{t \to \infty} \frac{\boldsymbol{w}(t)}{\|\boldsymbol{w}(t)\|} = \frac{\tilde{\boldsymbol{w}}}{\|\tilde{\boldsymbol{w}}\|},$$

*where $\tilde{\boldsymbol{w}} = \operatorname{argmin}_{\boldsymbol{w} \in \mathbb{R}^d} \|\boldsymbol{w}\|^2 \ s.t. \ \boldsymbol{w}^\top \boldsymbol{x}_i \ge 1, \ \forall \, i \in [n]$ denotes the max-margin solution.*

For simplicity, we denote $\boldsymbol{w}_+(t)$ as the parameter at iteration $t$ in the noiseless setting, and $\boldsymbol{w}_-(t)$ as the parameter in the noisy setting. By the proposition above, we know that both $\boldsymbol{w}_+(\infty)$ and $\boldsymbol{w}_-(\infty)$ are max-margin classifiers on the training data points. During the evaluation process, we also focus on the 0-1 loss, where the population 0-1 loss is $\mathcal{L}_{01}(\boldsymbol{w}) = \mathbb{P}(y\boldsymbol{x}^\top \boldsymbol{w} < 0)$.

## 3.3 MAIN THEOREM

Based on the above assumptions and discussions, we state the following Theorem 3.1, indicating the different performances between noiseless and noisy settings.

**Theorem 3.1.** *We consider the above GMM regime with Assumption [A1-A3]. Specifically, denote the noise level by $\rho$ and the mild overparameterization ratio by $r = p/n$. Then there exists absolute constant $c_1, c_2, c_3, c_4, c_5 > 0$ such that the following statements hold with probability[2] at least $1 - c_1/n$:*

1. *Under the noiseless setting, the max-margin classifier $\boldsymbol{w}_+(\infty)$ obtained from SGD has a non-vacuous 0-1 loss, namely,*

$$\mathcal{L}_{01}(\boldsymbol{w}_+(\infty)) \lesssim n^{-c_2}.$$

2. *Under the noisy setting, the max-margin classifier $\boldsymbol{w}_-(\infty)$ has vacuous 0-1 loss with a constant lower bound, namely, the following inequality holds for any training sample size $n$,*

$$\mathcal{L}_{01}(\boldsymbol{w}_-(\infty)) \ge \min\left\{\Phi(-2), \frac{\rho}{c_3 r} \exp\left(-\frac{c_3 r}{\rho}\right)\right\}.$$

3. *Under the noisy setting, if the learning rate satisfies $\eta < \frac{1}{c_5 n \max_i \|\boldsymbol{x}_i\|^2}$, there exists a time $t$ such that the trained early-stopping classifier $\boldsymbol{w}_-(t)$ has non-vacuous 0-1 loss, namely,*

$$\inf_{t \le n} \mathcal{L}_{01}(\boldsymbol{w}_-(t)) \lesssim n^{-c_4}.$$

Intuitively, Theorem 3.1 illustrates that although SGD leads to benign overfitting under noiseless regimes (statement 1), it provably overfits when the labels are noisy (statement 2). In particular, it incurs $\Omega(1)$ error on noiseless data and hence would incur $\Omega(1) + \rho$ error on noisy labeled data,

---

[2]The probability is taken over the training set and the randomness of the algorithm.

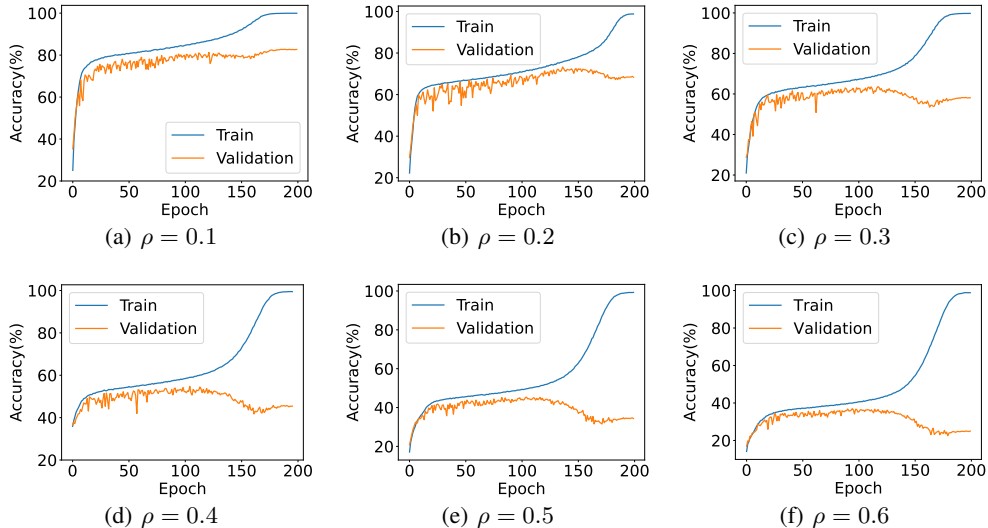

Figure 3: **Noisy CIFAR10 under mild overparameterization.** In each experiment, the validation accuracy first increases and then dramatically decreases. This confirms points 2 and 3 in Theorem 3.1 that model would overfit under noisy label regimes.

since label noise in the test set is independent of the algorithm. Furthermore, statement 3 shows that overfitting is avoidable through early stopping.

One may doubt the fast convergence rate[3] in Statement One and Statement Three, which seems too good to be true. The strange phenomenon happens because the high probability guarantee is in order $O(1/n)$ with respect to the randomness in the sampling of the training set and the algorithm, and therefore, the expected 0-1 loss is approximately $O(1/n)$ after union bound. We refer to Cao et al. (2021); Wang et al. (2021a) for similar types of bounds.

**Remark 1** (Comparison against heavy overparameterization). *Previous work usually analyze the GMM model under the* heavy overparameterization *regime,* e.g., $p = \Omega(n^2)$ *(Cao et al., 2021; Chatterji et al., 2021) or* $p = \Omega(n \log(n))$ *(Wang et al., 2021a). In comparison, our paper focuses on the mild overparameterization regime, where* $p = \Theta(n)$*. We note that this leads to a phase change that the overfitting model under noisy settings now provably overfits.*

## 4 EXPERIMENTS

Theorem 3.1 claims the crucial role of label noise and overparameterization level in benign overfitting. This section aims at providing experimental evidence to validate the statements. However, as the dataset size is upper bounded, the criterion *generalization error converges to zero* is hard to verify in experiments. Instead, we assume that (1) the gap of verification accuracy between the best-epoch model and the last-epoch model can work as an efficient proxy for the gap between the Bayesian optimal accuracy and the validation accuracy of the last-epoch model, and (2) training the model for long-enough epochs can work as an honest proxy for training for infinite epochs. Therefore, as stated in the introduction part, we adopt a different notion of benign overfitting, namely, *validation performance does not drop while the model fits more training data points.* Specifically, we focus on the noisy CIFAR10 dataset, where we randomly flip a portion of the labels. From the label noise perspective, Section 4.1 demonstrates that ResNets on noisy CIFAR10 would dramatically overfit non-benignly (*c.f.*, ResNets on clean CIFAR10 overfit benignly). From the overparameterization level perspective, Section 4.2 demonstrates that a heavier overparameterization level helps increase the last-iterate performance and hence leads to benign overfitting.

---

[3]Here the convergence rate is with respect to the sample size $n$, as stated in Theorem 3.1.

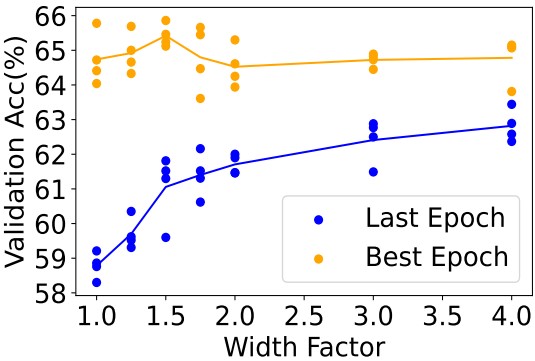 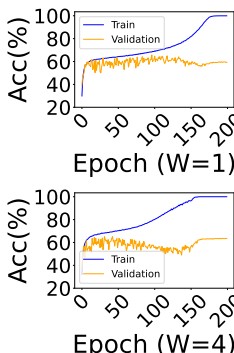

Figure 4: **Ablation Study on Overparameterization Level.** We fix the label noise ratio $\rho = 0.2$ and conduct an ablation study varying the width factor $W$ of WideResNet16-$W$. The left plot validates that generalization performances in the best iterate keep constant while the performances in the final iterate keep increasing, as the width factor increases. The training trajectories for width factors 1 and 4 are shown in the right two plots.

## 4.1 LABEL NOISE MATTERS: OVERFITTING IN NOISY CIFAR10

In Section 3.1, we prove that under noisy label regimes with mild overparameterization, early-stopping classifiers and interpolators perform differently. This section aims to verify if the phenomenon empirically happens in the real-world dataset. Specifically, we generate a noisy CIFAR10 dataset where each label is randomly flipped with probability $\rho$, We train the ResNets again and the results show that the test error first increases and then dramatically decreases, demonstrating that the interpolator performs worse than the models in the middle of the training.

**Setup.** The base dataset is CIFAR10, where each sample is randomly flipped with probability $\rho \in \{0.1, 0.2, 0.3, 0.4, 0.5, 0.6\}$. We use ResNet18 and SGD to train the model with cosine learning rate decay. For each model, we train for 200 epochs, test the validation accuracy and plot the training accuracy and validation accuracy in Figure 3. More details can be found in the code.

Figure 3 illustrates a similar phenomenon to the results in linear models under mild overparameterization analysis (see Section 3.1). Precisely, the interpolator achieves suboptimal accuracy under the mild overparameterization regimes, but we can still find a better classifier through early stopping. In Figure 3, such a phenomenon is manifested as the validation accuracy curve increases and decreases. We also notice that the degree becomes sharper as the noise level increases.

## 4.2 ACHIEVE BENIGN OVERFITTING BY INCREASING OVERPARAMETERIZATION LEVEL

Theorem 3.1 implies that although noisy label leads to bad overfitting, increasing overparameterization can turn overfitting benign. Specifically, when the overparameterization ratio $r = p/n$ increases,

(1) The last-iterate classifier (max-margin classifier) may generalize better (see Argument 2 in Theorem 3.1), since the lower bound $\frac{\rho}{c_3 r} \exp\left(-\frac{c_3 r}{\rho}\right)$ decreases with $r$.

(2) The early-stopping classifiers have similar generalization performances (see Argument 3 in Theorem 3.1), since the upper bound is in order $n^{-c_4}$ which converges to zero as the sample size goes to infinity.

Figure 2 demonstrates the two observations on the synthetic dataset (linear models). We next show similar observations on noisy CIFAR-10. Specifically, We conduct an ablation study on CIFAR10 dataset with a fixed label noise ratio $\rho = 0.2$. We increase the width of ResNets and effectively turn ResNet into Wide-ResNets. By varying the width factor $W$ of WideResNet16-$W$, we show that these observations remain in the deep learning regime.

**Setup.** The base dataset is CIFAR10, where each sample is randomly flipped with probability $\rho = 0.2$. We use the WideResNet16-$W$ for $W \in \{1, 1.25, 1.5, 1.75, 2, 3, 4\}$ and use SGD to train the model with cosine learning rate decay. For each model, we train for 200 epochs, test the validation accuracy and compare the best validation performance and the validation performance upon convergence in Figure 4.

Figure 4 verifies our theoretical observation where the early stopped classifier enjoys generalization performance nearly independent of overparameterization level, again stressing the necessity to study the bias of neural networks beyond the interpolation regime. Besides, we observe that the validation performance of the converged classifier increases with respect to the overparameterization level. Moreover, we notice that the increment is more significant for smaller overparameterization level, which is qualitatively similar to $\frac{\rho}{c_3 r} \exp(-\frac{c_3 r}{\rho})$.

We further observe the *epoch-wise double descent* phenomenon in Figure 4, where the accuracy first decreases and then increases while the model reaches interpolation (Nakkiran et al., 2020; Heckel et al., 2021). Stephenson et al. (2021) argues that the phenomenon is also closely related to label noise and overparameterization level. Their work differs from ours in that they focus on studying the training dynamics and removing the double-descent phenomenon. We do not observe a similar phenomenon in the ImageNet experiment with mild overparameterization in 500 epochs. We leave the discussion of epoch-wise double descent under mild overparameterization as future work.

## 5 CHALLENGES IN PROVING THEOREM 3.1

This section provides more details about the three statements in Theorem 3.1 with milder assumptions than those in Assumption 1. We also explain why existing analysis cannot be readily applied.

**Assumption 2.** *The following assumptions are more general,*

> *A4 The noise $\epsilon$ in $\boldsymbol{x}$ is generated from a $\sigma$-subGaussian distribution.*
>
> *A5 The signal-to-noise ratio satisfies $\frac{\|\boldsymbol{\mu}\|}{\sigma} \geq c_6 \left(\frac{p}{n}\right)^{\frac{1}{2}}$.*
>
> *A6 The signal-to-noise ratio satisfies $\frac{\|\boldsymbol{\mu}\|}{\sigma} = \omega\left(\left(\frac{p}{n}\right)^{\frac{1}{4}}\right)$.*

We compare the assumptions in Assumption 1 and Assumption 2. Assumption [A4] is a relaxation of Assumption [A1], and Assumption [A5, A6] can be obtained by Assumption [A2, A3]. Therefore, we conclude that Assumption 2 is weaker than Assumption 1. We next introduce the generalized version of the three arguments in Theorem 3.1, including Theorem 5.1, Theorem 5.2 and Theorem 5.3.

**Theorem 5.1** (Statement One). *Under the noiseless setting, for a fixed $\delta_1 \geq \max\{\frac{c_7}{n}, \exp(-c_8 p)\}$, under Assumption [A2, A4, A5], there exists constant $c_2 > 0$ such that the following statement holds with probability at least $1 - \delta_1$,*

$$\mathcal{L}_{01}(\boldsymbol{w}_+(\infty)) \lesssim n^{-c_2}.$$

Theorem 5.1 implies that the interpolator under noiseless regimes converges to zero as the sample size $n$ goes to infinity. The proof of Theorem 5.1 depends on bounding the projection of the classifier $\boldsymbol{w}$ on $\boldsymbol{\mu}$-direction, which relies on a sketch of the classification margin. We defer the whole proof to the Appendix due to space limitations.

Previous results on noiseless GMM (*e.g.*, Cao et al. (2021); Wang et al. (2021a)) rely on the heavy overparameterization $p = \omega(n)$ and assumption $\left(\frac{p}{n}\right)^{\frac{1}{4}} \leq \frac{\|\boldsymbol{\mu}\|}{\sigma} \leq \frac{p}{n}$. In contrast, our results only require $\frac{\|\boldsymbol{\mu}\|}{\sigma} \geq \left(\frac{p}{n}\right)^{\frac{1}{2}}$ as well as $\frac{\|\boldsymbol{\mu}\|}{\sigma} \geq c\sqrt{\log n}$, and therefore, can be deployed in the mild overparameterization regimes. Therefore, the existing results cannot directly imply Theorem 5.1.

**Theorem 5.2** (Statement Two). *Under the noisy regime with noisy level $\rho$ and mild overparameterization ratio $r = p/n$, for a fixed $\delta_2 \geq \max\{\exp(-c_8 p), c_9 \exp(-c_{10} \eta^2 n)\}$, under Assumption [A1, A3], there exists constant $c_3$ such that the following statement holds with probability at least $1 - \delta_2$,*

$$\mathcal{L}_{01}(\boldsymbol{w}_-(\infty)) \geq \min\left\{\Phi(-2), \frac{\rho}{c_3 r} \exp\left(-\frac{c_3 r}{\rho}\right)\right\}.$$

*Therefore, $\boldsymbol{w}_-(\infty)$ has constant excess risk, given that $r$ and $\rho$ are both constant.*

Theorem 5.2 proves a constant lower bound for interpolators in noisy settings. Compared to Theorem 5.1, Theorem 5.2 show that noisy and noiseless regimes perform differently under mild overparameterization. The core of the proof lies in controlling the distance between the center of wrong labeled samples and the point $\boldsymbol{\mu}$, which further leads to an upper bound of $|\boldsymbol{\mu}^\top \boldsymbol{w}_-(\infty)|$. One can then derive the corresponding 0-1 loss for classifier $\boldsymbol{w}_-(\infty)$. We defer the whole proof to the Appendix due to space limitations.

Previous results (*e.g.*, Chatterji et al. (2021); Wang et al. (2021a)) mainly focus on deriving the non-vacuous bound for noisy GMM, which also relies on the heavy overparameterization assumption $p = \omega(n)$. Instead, our results show that the interpolator dramatically fails and suffers from a constant lower bound under mild overparameterization regimes. Therefore, heavy overparameterization performs differently from mild overparameterization cases. We finally remark that although it is still an open problem when the phase change happens, we conjecture that realistic training procedures are more close to the mild overparameterization regime according to the experiment results.

**Theorem 5.3** (Statement Three). *Under the noisy regime, consider the learning rate* $\eta < \frac{1}{c_5 n \max_i \|\boldsymbol{x}_i\|^2}$ *where* $(\boldsymbol{x}_i, \tilde{y}_i) \in \mathcal{D}$ *denotes the data point. For a fixed* $\delta_3 \geq \max\{\frac{c_{11}}{n}, c_{12} \exp(-c_{13}\eta^2 n)\}$, *under Assumption [A2, A4, A6], the following statement holds with probability at least* $1 - \delta_3$,

$$\inf_{t \leq n} \mathcal{L}_{01}(\boldsymbol{w}_-(t)) \leq \exp\left(-c_{14}\frac{\|\boldsymbol{\mu}\|^4}{\|\boldsymbol{\mu}\|^2 \sigma^2 + \sigma^4 \frac{p}{n}}\right).$$

*Therefore, the bound converges to zero as sample size $n$ goes to infinity under Assumption [A6].*

Theorem 5.3 derives that the bound again converges to zero by considering early-stopping. Studying early-stopping classifiers is meaningful since people usually cannot ideally obtain the interpolation during training in practice. The derivation of Theorem 5.3 relies on the analysis of one-pass SGD. Different from the previous approaches where we can directly assume $\|\boldsymbol{w}_+(\infty)\| = \|\boldsymbol{w}_-(\infty)\| = 1$ without loss of generality, the classifier $\boldsymbol{w}_-(t)$ is trained in this case and we need to first bound it. We then define a surrogate classifier and show that (a) the surrogate classifier is close to the trained classifier for a sufficiently small learning rate, and (b) the surrogate classifier can return a satisfying projection on the direction $\boldsymbol{\mu}$. Therefore, we bound the term $\boldsymbol{\mu}^\top \boldsymbol{w}_-(t)/\|\boldsymbol{w}_-(t)\|$ which leads to the results. We defer the whole proof to the Appendix due to space limitations.

One may wonder whether we can apply the results of stability-based bound (Bousquet et al., 2002; Hardt et al., 2016) into the analysis since the training process is convex. However, the analysis might not be proper due to a bad Lipschitz constant during the training process. Therefore, the stability-based analysis may only return vacuous bound under such regimes. Besides, the previous results on convex optimization with one-pass SGD (*e.g.*, Sekhari et al. (2021)) cannot be directly applied to the analysis since most results on one-pass SGD are expectation bounds, while we provide a high probability bound in Theorem 5.3.

## 6 CONCLUSIONS AND DISCUSSIONS

In this work, we aim to understand why benign overfitting happens in training ResNet on CIFAR10, but fails on ImageNet. We start by identifying a phase change in the theoretical analysis of benign overfitting. We found that when the model parameter is in the same order as the number of data points, benign overfitting would fail due to label noise. We conjecture that the noise in labels leads to the different behaviors in CIFAR10 and ImageNet. We verify the conjecture by injecting label noise into CIFAR10 and adopting self-training in ImageNet. The results support our hypothesis.

Our work also left many questions unanswered. First, our theoretical and empirical evidence shows that realistic deep learning models may not work in the interpolating scheme. Still, although there is a larger number of parameters than data points, the model generalizes well. Understanding the implicit bias in deep learning when the model underfits is still open. A closely related topic would be algorithmic stability (Bousquet et al., 2002; Hardt et al., 2016), however, the benefit of overparameterization within the stability framework still requires future studies. Second, the GMM model provides a convenient way for analysis, but how the number of parameters in the linear setup relates to that in the neural network remains unclear. Third, although the conclusions in this paper can be generalized to other regimes, e.g., GMM with Gradient Descent, the possible extension to neural networks as in Cao et al. (2022) is still unclear and we will consider that in future works.

ACKNOWLEDGMENTS

Jingzhao Zhang acknowledges support by Tsinghua University Initiative Scientific Research Program.

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

# Appendix

## A  ADDITIONAL RELATED WORKS ON LABEL NOISE

Label noise often exists in real world datasets, *e.g.*, in ImageNet (Yun et al., 2021; Shankar et al., 2020). However, its effect on generalization remains debatable. Recently, Damian et al. (2021) claim that in regression settings, label noise may prefer flat global minimizers and thus helps generalization. Another line of work claims that label noise hurts the effectiveness of empirical risk minimization in classification regimes, and proposes to improve the model performance by explicit regularization (Mignacco et al., 2020) or robust training (Brodley et al., 1996; Guan et al., 2011; Huang et al., 2020). Among them, Bagherinezhad et al. (2018) studies how to refine the labels in ImageNet using label propagation to improve the model performance, demonstrating the importance of the label in ImageNet. Besides, Nakkiran et al. (2021a) find that interpolation does harm to generalization in the presence of label noise. However, they do not relate their findings with the scale of overparameterization. This paper mainly falls in the latter branch, which theoretically analyzes how label noise acts in the mild overparameterization regimes.

## B  DETAILED PROOFS

### B.1  PROOF OF THEOREM 5.1

The first statement in Theorem 3.1 states that the interpolator has a non-vacuous bound in a noiseless setting, which is a direct corollary of the following Theorem 5.1. The proof of Theorem 5.1 mainly depends on bounding the projection of the classifier $\boldsymbol{w}$ on $\boldsymbol{\mu}$, which relies on a sketch of the classification margin.

**Theorem 5.1** (Statement One). *Under the noiseless setting, for a fixed $\delta_1 \geq \max\{\frac{c_7}{n}, \exp(-c_8 p)\}$, under Assumption [A2, A4, A5], there exists constant $c_2 > 0$ such that the following statement holds with probability at least $1 - \delta_1$,*
$$\mathcal{L}_{01}(\boldsymbol{w}_+(\infty)) \lesssim n^{-c_2}.$$

*Proof of Theorem 5.1.* We denote the classifier $\boldsymbol{w}_+(\infty)$ by $\boldsymbol{w}$ during the proof for simplicity. Due to Proposition 3.1, the final classifier converges to its max-margin solution. Without loss of generality, let the final classifier $\boldsymbol{w}$ satisfy $\|\boldsymbol{w}\| = 1$. Therefore, the following equation for margin $\gamma(\cdot)$ holds since $\boldsymbol{w}$ is max-margin solution:
$$\gamma(\boldsymbol{w}) \geq \gamma(\boldsymbol{\mu}/\|\boldsymbol{\mu}\|),$$
where $\gamma(\boldsymbol{w}) = \min_i y_i \boldsymbol{x}_i^\top \boldsymbol{w}$ denotes the margin of classifier $\boldsymbol{w}$ for the dataset. We next consider the margin for the classifier $\boldsymbol{\mu}/\|\boldsymbol{\mu}\|$. Note that the margin can be rewritten as
$$
\begin{aligned}
&\gamma(\boldsymbol{\mu}/\|\boldsymbol{\mu}\|) \\
=& \min_i y_i \boldsymbol{x}_i^\top \boldsymbol{\mu}/\|\boldsymbol{\mu}\| \\
=& \min_i y_i (y_i \boldsymbol{\mu}^\top + \boldsymbol{\epsilon}_i^\top) \boldsymbol{\mu}/\|\boldsymbol{\mu}\| \\
=& \|\boldsymbol{\mu}\| + \min_i y_i \boldsymbol{\epsilon}_i^\top \boldsymbol{\mu}/\|\boldsymbol{\mu}\| \\
\geq& \|\boldsymbol{\mu}\| - \max_i |y_i \boldsymbol{\epsilon}_i^\top \boldsymbol{\mu}/\|\boldsymbol{\mu}\||.
\end{aligned}
$$

We note that $y_i \boldsymbol{\epsilon}_i^\top \boldsymbol{\mu}/\|\boldsymbol{\mu}\|$ is $\sigma$-subGaussian due to the definition of subGaussian random vector. Therefore, due to Claim C.1, we have
$$\gamma(\boldsymbol{w}) \geq \gamma(\boldsymbol{\mu}/\|\boldsymbol{\mu}\|) \gtrsim \|\boldsymbol{\mu}\| - \sigma\sqrt{\log n} \gtrsim \|\boldsymbol{\mu}\|,$$
where the last inequality is due to Assumption [A2].

We next bound the term $\boldsymbol{\mu}^\top \boldsymbol{w}$ via the above margin. From one hand, we notice that by the definition of the margin function,
$$\gamma(\boldsymbol{w}) = \min_i y_i \boldsymbol{x}_i^\top \boldsymbol{w} \gtrsim \|\boldsymbol{\mu}\|. \tag{1}$$

From the other hand, we rewrite the margin as

$$
\begin{aligned}
&\min_i y_i \boldsymbol{x}_i^\top \boldsymbol{w} \\
=& \boldsymbol{\mu}^\top \boldsymbol{w} + \min_i y_i \boldsymbol{\epsilon}_i^\top \boldsymbol{w} \\
\leq& \boldsymbol{\mu}^\top \boldsymbol{w} + \frac{1}{n} \sum_{i \in [n]} y_i \boldsymbol{\epsilon}_i^\top \boldsymbol{w}.
\end{aligned}
\tag{2}
$$

The right hand side can be bounded as

$$
\begin{aligned}
&\frac{1}{n} \sum_{i \in [n]} y_i \boldsymbol{\epsilon}_i^\top \boldsymbol{w} \\
\leq& \left\| \frac{1}{n} \sum_{i \in [n]} y_i \boldsymbol{\epsilon}_i \right\| \\
\lesssim& \sigma \sqrt{\frac{p}{n}},
\end{aligned}
\tag{3}
$$

where the final equation is due to Claim C.2. Therefore, combining the above Equation 1, Equation 2 and Equation 3, we bound the projection of $\boldsymbol{w}_+$ on $\mu$ as:

$$
\boldsymbol{\mu}^\top \boldsymbol{w} \gtrsim \|\boldsymbol{\mu}\| - \sigma \sqrt{\frac{p}{n}} \gtrsim \|\boldsymbol{\mu}\|,
\tag{4}
$$

where the last equation is due to Assumption [A5]. We rewrite Equation (4) as $\boldsymbol{\mu}^\top \boldsymbol{w} \geq c_5 \|\boldsymbol{\mu}\|$, then we can bound the 0-1 loss as follows for a given constant $c_6$:

$$
\begin{aligned}
\mathcal{L}_{01}(\boldsymbol{w}) =& \mathbb{P}(y \boldsymbol{x}^\top \boldsymbol{w} < 0) \\
=& \mathbb{P}(\boldsymbol{\mu}^\top \boldsymbol{w} + y_i \boldsymbol{\epsilon}^\top \boldsymbol{w} < 0) \\
=& \mathbb{P}(y \boldsymbol{\epsilon}^\top \boldsymbol{w} \leq -\boldsymbol{\mu}^\top \boldsymbol{w}) \\
\leq& \mathbb{P}(y \boldsymbol{\epsilon}^\top \boldsymbol{w} \leq -c_5 \|\boldsymbol{\mu}\|) \\
\leq& \exp\left(-c_6 \frac{\|\boldsymbol{\mu}\|^2}{\sigma^2}\right).
\end{aligned}
$$

Due to Assumption [A2], $\|\boldsymbol{\mu}\|/\sigma > c\sqrt{\log n}$, and therefore, by setting $c_2 = c_6 c^2$, we have

$$
\mathcal{L}_{01}(\boldsymbol{w}) \lesssim n^{-c_2}.
$$

The proof is done. $\qquad\square$

## B.2  PROOF OF THEOREM 5.2

Theorem 5.1 shows that the interpolator can be non-vacuous under noiseless regimes with mild overparameterization. However, things can be much different in noisy regimes. Statement Two proves a vacuous lower bound for interpolators in noisy settings, which can be derived by the following Theorem 5.2. The core of the proof lies in controlling the distance between the center of wrong labeled samples and the point $\boldsymbol{\mu}$, which further leads to an upper bound of $|\boldsymbol{\mu}^\top \boldsymbol{w}_-(\infty)|$. One can then derive the corresponding 0-1 loss for classifier $\boldsymbol{w}_-(\infty)$.

**Theorem 5.2** (Statement Two). *Under the noisy regime with noisy level $\rho$ and mild overparameterization ratio $r = p/n$, for a fixed $\delta_2 \geq \max\{\exp(-c_8 p), c_9 \exp(-c_{10}\eta^2 n)\}$, under Assumption [A1, A3], there exists constant $c_3$ such that the following statement holds with probability at least $1 - \delta_2$,*

$$
\mathcal{L}_{01}(\boldsymbol{w}_-(\infty)) \geq \min\left\{\Phi(-2), \frac{\rho}{c_3 r}\exp\left(-\frac{c_3 r}{\rho}\right)\right\}.
$$

*Therefore, $\boldsymbol{w}_-(\infty)$ has constant excess risk, given that $r$ and $\rho$ are both constant.*

*Proof.* We denote $\boldsymbol{w}_-(\infty)$ as $\boldsymbol{w}$ for simplicity, and assume that $\|\boldsymbol{w}\| = 1$ without loss of generality. Let $y$ denote the original label and $\tilde{y}$ denote its corrupted label.

Without loss of generality, we consider those samples with $y_i = 1$ while $\tilde{y}_i = -1$, which are indexed by $\mathcal{K} = \{i : y_i = 1, \tilde{y}_i = -1\}$. Consider the center point of $\mathcal{K}$, which is

$$\bar{\boldsymbol{x}}_{\mathcal{K}} = \frac{1}{|\mathcal{K}|} \sum_{i \in \mathcal{K}} \boldsymbol{x}_i = \boldsymbol{\mu} \frac{1}{|\mathcal{K}|} \sum_{i \in \mathcal{K}} y_i + \frac{1}{|\mathcal{K}|} \sum_{i \in \mathcal{K}} \boldsymbol{\epsilon}_i = \boldsymbol{\mu} + \frac{1}{|\mathcal{K}|} \sum_{i \in \mathcal{K}} \boldsymbol{\epsilon}_i.$$

Due to the interpolation in Proposition 3.1 and $\tilde{y} = -1$, we derive that

$$\bar{\boldsymbol{x}}_{\mathcal{K}}^\top \boldsymbol{w} < 0. \tag{5}$$

**Case 1: $\boldsymbol{\mu}^\top \boldsymbol{w} < 0$.** In this case, $\mathcal{L}_{01}(\boldsymbol{w})$ naturally has a lower bound of $1/2$ since it even fails in the center point $\boldsymbol{\mu}$.

**Case 2: $\boldsymbol{\mu}^\top \boldsymbol{w} > 0$.** In this case, the classifier $\boldsymbol{w}$ satisfies $\boldsymbol{\mu}^\top \boldsymbol{w} > 0$ and $\bar{\boldsymbol{x}}_{\mathcal{K}}^\top \boldsymbol{w} < 0$. Therefore, the distance between $\boldsymbol{\mu}$ and $\bar{\boldsymbol{x}}_{\mathcal{K}}^\top$ must be less than the distance from $\boldsymbol{\mu}$ to its projection on the separating hyperplane that perpendicular to $\boldsymbol{w}$ through the origin. Formally,

$$|\boldsymbol{\mu}^\top \boldsymbol{w}| \leq \|\bar{\boldsymbol{x}}_{\mathcal{K}}^\top - \boldsymbol{\mu}\| = \frac{1}{|\mathcal{K}|} \left\| \sum_{i \in \mathcal{K}} \boldsymbol{\epsilon}_i \right\|.$$

Note that $\boldsymbol{\epsilon}_i$ is independent and $\sigma$-subGaussian, and therefore applying Claim C.2, we have that $\left\| \sum_{i \in \mathcal{K}} \boldsymbol{\epsilon}_i \right\| \lesssim \sigma \sqrt{\frac{p}{|\mathcal{K}|}}$. Besides, we derive by Claim C.3 that $|\mathcal{K}| \gtrsim \rho n$. Therefore,

$$|\boldsymbol{\mu}^\top \boldsymbol{w}| \leq \left\| \frac{1}{|\mathcal{K}|} \sum_{i \in \mathcal{K}} \boldsymbol{\epsilon}_i \right\| \lesssim \sigma \sqrt{\frac{d}{|\mathcal{K}|}} \lesssim \sigma \sqrt{\frac{p}{\rho n}}. \tag{6}$$

We next consider the corresponding test error of $\boldsymbol{w}$, where we consider the test error on noiseless regime instead of noisy regime. Note that the two arguments are equivalent, we refer to Claim C.4 for more details. We rewrite Equation 6 as $|\boldsymbol{\mu}^\top \boldsymbol{w}| \leq c\sigma \sqrt{\frac{d}{|\mathcal{K}|}} \lesssim \sigma \sqrt{\frac{p}{\rho n}}$, where we abuse the notation $c > 0$ as a fixed constant. Therefore,

$$
\begin{aligned}
&\mathbb{P}(y\boldsymbol{x}^\top \boldsymbol{w} < 0) \\
=&\mathbb{P}(y\boldsymbol{\epsilon}^\top \boldsymbol{w} < -\boldsymbol{\mu}^\top \boldsymbol{w}) \\
\geq&\mathbb{P}(y\boldsymbol{\epsilon}^\top \boldsymbol{w}/\sigma < -c\sqrt{\frac{p}{\rho n}}) \\
=&\Phi(-c\sqrt{\frac{p}{\rho n}}),
\end{aligned}
\tag{7}
$$

where $\boldsymbol{\epsilon}$ is sampled from Gaussian distribution, and $\Phi$ denotes the CDF of standard Gaussian Random Variable.

*Case 1.* If $c\sqrt{\frac{p}{\rho n}} \leq 2$, then $\Phi(-c\sqrt{\frac{p}{\rho n}}) \geq \Phi(-2)$.

*Case 2.* If $c\sqrt{\frac{p}{\rho n}} > 2$, note that $\Phi(-t) \geq (\frac{1}{t} - \frac{1}{t^3}) \exp(-t^2/2) \geq 1/t^2 \exp(-t^2/2)$ if $t > 2$. Therefore, $\Phi(-c\sqrt{\frac{p}{\rho n}}) \geq \frac{1}{c^2 \frac{p}{\rho n}} \exp(-\frac{c^2}{2} \frac{p}{\rho n}) \geq \frac{1}{c^2 \frac{p}{\rho n}} \exp(-c^2 \frac{p}{\rho n})$.

Taking the above two cases together and denoting $r = p/n$, we have

$$\mathcal{L}_{01}(\boldsymbol{w}) \geq \min \left\{ \Phi(-2), \frac{\rho}{c_3 r} \exp(-\frac{c_3 r}{\rho}) \right\},$$

which is a constant lower bound under Assumption [A3]. The proof is done. $\qquad\square$

### B.3 PROOF OF THEOREM 5.3

We apologize for the following typos made in Theorem 5.3 in the main text. For consistency, the dimension $d$ should be written as $p$ and the learning rate $\lambda$ should be written as $\eta$. Furthermore, Theorem 5.3 needs an initialization from zero during the training, which is widely used in linear models (Bartlett et al., 2020).

Statement Two shows that the interpolator fails in the noisy regime with mild overparameterization. How can we derive a non-vacuous bound under such regimes? The key is early-stopping. To show that, Statement Three provides a non-vacuous bound for early-stopping classifiers in noisy regimes, which is induced by the following Theorem 5.3.

**Theorem 5.3** (Statement Three). *Under the noisy regime, consider the learning rate* $\eta < \frac{1}{c_5 n \max_i \|\boldsymbol{x}_i\|^2}$ *where* $(\boldsymbol{x}_i, \tilde{y}_i) \in \mathcal{D}$ *denotes the data point. For a fixed* $\delta_3 \geq \max\{\frac{c_{11}}{n}, c_{12} \exp(-c_{13}\eta^2 n)\}$, *under Assumption [A2, A4, A6], the following statement holds with probability at least* $1 - \delta_3$,

$$\inf_{t \leq n} \mathcal{L}_{01}(\boldsymbol{w}_-(t)) \leq \exp\left(-c_{14} \frac{\|\boldsymbol{\mu}\|^4}{\|\boldsymbol{\mu}\|^2 \sigma^2 + \sigma^4 \frac{p}{n}}\right).$$

*Therefore, the bound converges to zero as sample size $n$ goes to infinity under Assumption [A6].*

To show the relationship between Theorem 5.3 and Theorem 3.1, one can directly use Assumption [A2, A3] in Theorem 5.3 to reach generalization bound in Theorem 3.1 (Statement Three). The derivation of Theorem 5.3 relies on the analysis on one-pass SGD, where we show that one-pass SGD is sufficient to reach non-vacuous bound. The proof of Theorem 5.3 again, relies on bounding the term c but in a different way. Different from the previous approaches where we can directly assume $\|\boldsymbol{w}_+(\infty)\| = \|\boldsymbol{w}_-(\infty)\| = 1$, the classifier $\boldsymbol{w}_-(t)$ is trained in this case and we need to first bound it. We then define a surrogate classifier and show that (a) the surrogate classifier is close to the trained classifier for a sufficiently small learning rate, and (b) the surrogate classifier can return satisfying projection on the direction $\boldsymbol{\mu}$. Therefore, we bound the term $\boldsymbol{\mu}^\top \boldsymbol{w}_-(t)/\|\boldsymbol{w}_-(t)\|$ which leads to the results.

*Proof.* We abuse the notation $\boldsymbol{w}_n$ to represent $\boldsymbol{w}_-(t)$ which is returned by one-pass SGD. We first lower bound the term $\frac{\boldsymbol{\mu}^\top \boldsymbol{w}_n}{\|\boldsymbol{w}_n\|}$, where $\mu$ is the optimal classification direction. To achieve the goal, we bound the term $\boldsymbol{\mu}^\top \boldsymbol{w}_n$ and $\|\boldsymbol{w}_n\|$ individually.

Before diving into the proof, we first introduce a surrogate classifier $\tilde{\boldsymbol{w}}_n = \frac{1}{2}\eta \sum_t \boldsymbol{x}_t \tilde{y}_t$. From the definition, we have that for update step size $\eta$,

$$\boldsymbol{w}_{t+1} = \boldsymbol{w}_t + \eta \boldsymbol{x}_t \tilde{y}_t \frac{\exp(-y_i \boldsymbol{x}_i \boldsymbol{w}_t)}{1 + \exp(-y_i \boldsymbol{x}_i \boldsymbol{w}_t)},$$

$$\tilde{\boldsymbol{w}}_{t+1} = \tilde{\boldsymbol{w}}_t + \frac{1}{2}\eta \boldsymbol{x}_t \tilde{y}_t. \tag{8}$$

Therefore, we have that

$$\boldsymbol{w}_{t+1} - \tilde{\boldsymbol{w}}_{t+1} = \eta \sum_t \boldsymbol{x}_t \tilde{y}_t \frac{\exp(-\tilde{y}_t \boldsymbol{x}_t^\top \boldsymbol{w}_t) - 1}{2(1 + \exp(-\tilde{y}_t \boldsymbol{x}_t^\top \boldsymbol{w}_t))}.$$

$$\boldsymbol{\mu}^\top \boldsymbol{w}_{t+1} - \boldsymbol{\mu}^\top \tilde{\boldsymbol{w}}_{t+1} = \eta \sum_t \boldsymbol{\mu}^\top \boldsymbol{x}_t \tilde{y}_t \frac{\exp(-\tilde{y}_t \boldsymbol{x}_t^\top \boldsymbol{w}_t) - 1}{2(1 + \exp(-\tilde{y}_t \boldsymbol{x}_t^\top \boldsymbol{w}_t))} \tag{9}$$

*Bounding the term $\boldsymbol{\mu}^\top \boldsymbol{w}$.* We fist bound the different between $\boldsymbol{\mu}^\top \boldsymbol{w}_t$ and $\boldsymbol{\mu}^\top \tilde{\boldsymbol{w}}_t$. We note that

To bound the above different, the fist step is to bound the term $\max_{t \in [n]} \boldsymbol{\mu}^\top \boldsymbol{x}_t \tilde{y}_t$. Since $\boldsymbol{\mu}^\top \boldsymbol{\epsilon}_t/\|\boldsymbol{\mu}\|$ is $\sigma$-subGuassian, we have that with probability $1 - \delta_1$ with $\delta_1 \lesssim 1/n$

$$\max_{t \in [n]} |\boldsymbol{\mu}^\top \boldsymbol{x}_t \tilde{y}_t| \leq \|\boldsymbol{\mu}\|^2 + \max_t |\boldsymbol{\mu}^\top \boldsymbol{\epsilon}_t| \lesssim \|\boldsymbol{\mu}\|^2 + \|\boldsymbol{\mu}\|\sigma\sqrt{\log(n/\delta_1)} \lesssim \|\boldsymbol{\mu}\|^2 + \|\boldsymbol{\mu}\|\sigma\sqrt{\log(n)} \lesssim \|\boldsymbol{\mu}\|^2.$$
$$\tag{10}$$

We then bound the term $\frac{\exp(-\tilde{y}_t \boldsymbol{x}_t^\top \boldsymbol{w}_t)-1}{2(1+\exp(-\tilde{y}_t \boldsymbol{x}_t^\top \boldsymbol{w}_t))}$. Note that $|\exp(u) - 1| \le 2|u|$ when $|u| < \frac{1}{2}$. By the iteration in Equation 8, we have that

$$\max_{t \in [n]} \|\boldsymbol{w}_t\| \le n\eta(\max_t \|\boldsymbol{x}_t\|),$$

$$\max_{t \in [n]} |\boldsymbol{x}_t^\top \boldsymbol{w}_t| \le n\eta(\max_t \|\boldsymbol{x}_t\|^2) \le \frac{1}{2}.$$

where the first equation is due to the iteration, and the last equation is due to $\eta \le \frac{1}{2n \max_i \|\boldsymbol{x}_i\|^2}$ (by setting $c_5 > 2$). Therefore,

$$\left| \frac{\exp(-\tilde{y}_t \boldsymbol{x}_t^\top \boldsymbol{w}_t) - 1}{2(1 + \exp(-\tilde{y}_t \boldsymbol{x}_t^\top \boldsymbol{w}_t))} \right| \le \left| \frac{\exp(-\tilde{y}_t \boldsymbol{x}_t^\top \boldsymbol{w}_t) - 1}{2} \right| \le \left| \boldsymbol{x}_t^\top \boldsymbol{w}_t \right| \le \frac{1}{2}. \tag{11}$$

Combining Equation 10 and Equation 11, we have that with probability at least $1 - \delta_1$,

$$\max_{t \in [n]} |\boldsymbol{\mu}^\top \boldsymbol{w}_t - \boldsymbol{\mu}^\top \tilde{\boldsymbol{w}}_t| \lesssim \eta n \|\boldsymbol{\mu}\|^2. \tag{12}$$

On the other hand, we show the bound for $\boldsymbol{\mu}^\top \tilde{w}_n$. Note that $\tilde{w}_{t+1} = \eta/2 \sum_t \boldsymbol{x}_t \tilde{y}_t$, therefore,

$$\frac{2}{n\eta} \boldsymbol{\mu}^\top \tilde{\boldsymbol{w}}_n = \|\boldsymbol{\mu}\|^2 \frac{1}{n} \sum_{t \in [n]} \mathbb{I}_i(\rho) + \frac{1}{n} \sum_{t \in [n]} \boldsymbol{\mu}^\top \boldsymbol{\epsilon}_t \tilde{y}_t,$$

where we denote $\mathbb{I}_i(\rho) \in \{-1, 1\}$ as a random variable which takes value $-1$ with probability $\rho$ and takes value $+1$ with probability $1 - \rho$.

Due to Claim C.3, we have $\frac{1}{n} \sum_t \mathbb{I}_i(\rho) \gtrsim \rho$, where we note that $\mathbb{I}(\rho) \in \{-1, 1\}$. Besides, since $\boldsymbol{\mu}^\top \boldsymbol{\epsilon}_t \tilde{y}_t$ is $\|\boldsymbol{\mu}\|\sigma$-subGaussian, we have that with probability $1 - \frac{1}{n}$,

$$\frac{1}{n} \sum_{t \in [n]} \boldsymbol{\mu}^\top \boldsymbol{\epsilon}_t \tilde{y}_t \lesssim \|\boldsymbol{\mu}\|\sigma \sqrt{\log(n)},$$

where the term $\log(n)$ comes from the probability $1/n$. In summary, we have that

$$\frac{2}{n\eta} \boldsymbol{\mu}^\top \tilde{\boldsymbol{w}}_n \gtrsim \rho\|\boldsymbol{\mu}\|^2 - \|\boldsymbol{\mu}\|\sigma\sqrt{\log(n)} \gtrsim \|\boldsymbol{\mu}\|^2, \tag{13}$$

where the last equation follows Assumption [A2].

Combining Equation (12) and Equation (13), we have

$$\boldsymbol{\mu}^\top \boldsymbol{w}_n \ge \boldsymbol{\mu}^\top \tilde{\boldsymbol{w}}_n - |\boldsymbol{\mu}^\top \tilde{\boldsymbol{w}}_n - \boldsymbol{\mu}^\top \boldsymbol{w}_n| \gtrsim \eta n \|\boldsymbol{\mu}\|^2. \tag{14}$$

We additionally note that Equation 14 holds by choosing proper constant in Equation (11) (and in the choice of $\eta$).

*Bounding the norm $\|\boldsymbol{w}_n\|$.* We next bound the norm $\|\boldsymbol{w}_n\|$. Before that, we first bound the norm $\|\tilde{\boldsymbol{w}}_t\|$. Note that

$$\begin{aligned}
\max_{t \in [T]} \frac{2}{\eta} \|\tilde{\boldsymbol{w}}_t\| &= \max_{t \in [T]} \| \sum_{i \in [t]} \boldsymbol{x}_i \tilde{y}_i \| \\
&= \max_{t \in [T]} \| \boldsymbol{\mu}^\top \sum_{i \in [t]} \mathbb{I}_i(\rho) + \sum_{i \in [t]} \boldsymbol{\epsilon}_i \tilde{y}_i \| \\
&\le \max_{t \in [T]} \| \boldsymbol{\mu}^\top \sum_{i \in [t]} \mathbb{I}_i(\rho) \| + \| \sum_{i \in [t]} \boldsymbol{\epsilon}_i \tilde{y}_i \| \\
&\lesssim T\|\boldsymbol{\mu}\| + \sigma\sqrt{pT},
\end{aligned} \tag{15}$$

where the last equation is due to Claim C.2 by choosing probability $\delta \gtrsim \exp(-p)/n$.

Therefore, we have

$$\sup_{t \in [n]} \|\tilde{\boldsymbol{w}}_t\| \lesssim \eta n \|\boldsymbol{\mu}\| + \eta \sigma \sqrt{pn}.$$

Note that according to the iteration in Equation 8, we have that

$$\|\boldsymbol{w}_{t+1} - \tilde{\boldsymbol{w}}_{t+1}\| \le \eta t \max_i \|\boldsymbol{x}_i\|^2 \|\boldsymbol{w}_t\| \le \frac{1}{2} \|\boldsymbol{w}_t\|,$$

where the last equation is due to $\eta \le \frac{1}{2n \max_i \|\boldsymbol{x}_i\|^2}$.

Therefore, we bound the norm $\boldsymbol{w}_t$ as

$$
\begin{aligned}
\|\boldsymbol{w}_n\| &\le \|\boldsymbol{w}_n - \tilde{\boldsymbol{w}}_n\| + \|\tilde{\boldsymbol{w}}_n\| \le \frac{1}{2}\|\boldsymbol{w}_{n-1}\| + \|\tilde{\boldsymbol{w}}_n\| \\
&\le \frac{1}{4}\|\boldsymbol{w}_{n-2}\| + \frac{1}{2}\|\tilde{\boldsymbol{w}}_{n-1}\| + \|\tilde{\boldsymbol{w}}_n\| \\
&\le \dots \\
&\le 2\eta n \|\boldsymbol{\mu}\| + \eta\sigma\sqrt{pn} \\
&\lesssim \eta n \|\boldsymbol{\mu}\| + \eta\sigma\sqrt{pn}.
\end{aligned}
\tag{16}
$$

Combining Equation 14 and Equation 16, we derive that with probability at least $1 - c/n$,

$$\frac{\boldsymbol{\mu}^\top \boldsymbol{w}_n}{\|\boldsymbol{w}_n\|\sigma} \ge \frac{\eta n \|\boldsymbol{\mu}\|^2}{\eta n \sigma \|\boldsymbol{\mu}\| + \eta \sigma^2 \sqrt{pn}}.\tag{17}$$

We next consider the probability on the test point, note that given the dataset $(\boldsymbol{x}_i, y_i)$ and taking probability on the testing point $(\boldsymbol{x}, y)$, we have

$$\mathbb{P}(y\boldsymbol{x}^\top \boldsymbol{w} < 0) = \mathbb{P}(y\boldsymbol{\epsilon}^\top \boldsymbol{w}/\|\boldsymbol{w}\| \le -\boldsymbol{\mu}^\top \boldsymbol{w}/\|\boldsymbol{w}\|) \le \exp\left(-c\frac{(\boldsymbol{\mu}^\top \boldsymbol{w})^2}{\|\boldsymbol{w}\|^2\sigma^2}\right),$$

where $y\boldsymbol{x}^\top \boldsymbol{w} = \boldsymbol{\mu}^\top \boldsymbol{w} + y\boldsymbol{\epsilon}^\top \boldsymbol{w}$ and $y\boldsymbol{\epsilon}^\top \boldsymbol{w}/\|\boldsymbol{w}\|$ is $\sigma$-subGaussian. Plugging Equation 17 into the above equation, we have that with high probability,

$$\mathbb{P}(y\boldsymbol{x}^\top \boldsymbol{w} < 0) \le \exp\left(-c_{14}\frac{\|\boldsymbol{\mu}\|^4}{\|\boldsymbol{\mu}\|^2\sigma^2 + \sigma^4 \frac{p}{n}}\right).$$

If $\|\boldsymbol{\mu}\|/\sigma > \sqrt{p/n}$, $\frac{\|\boldsymbol{\mu}\|^4}{\|\boldsymbol{\mu}\|^2\sigma^2 + \sigma^4 \frac{p}{n}} \gtrsim \|\boldsymbol{\mu}\|/\sigma$.

If $\|\boldsymbol{\mu}\|/\sigma < \sqrt{p/n}$, $\frac{\|\boldsymbol{\mu}\|^4}{\|\boldsymbol{\mu}\|^2\sigma^2 + \sigma^4 \frac{p}{n}} \gtrsim \frac{\|\boldsymbol{\mu}\|^2}{\sigma^2}\sqrt{n/p}$, which is large when $\|\boldsymbol{\mu}\|/\sigma = \omega((p/n)^{1/4})$.

Therefore, the above bound is non-vacuous if $\|\boldsymbol{\mu}\|/\sigma = \omega((p/n)^{1/4})$.

Note that for given a constant $r = p/n$, under Assumption [A2], we have

$$\mathbb{P}(y\boldsymbol{x}^\top \boldsymbol{w} < 0) \le \exp\left(-c_{14}\frac{\|\boldsymbol{\mu}\|^4}{\|\boldsymbol{\mu}\|^2\sigma^2 + \sigma^4 \frac{p}{n}}\right) \lesssim \exp\left(-c_{14}\frac{\|\boldsymbol{\mu}\|^2}{\sigma^2}\right) \lesssim n^{-c_4}.$$

The proof is done. $\qquad\qquad\square$

## C  TECHNICAL LEMMAS

We next provide some technical claims. The first claims bound the maximum of a sequence of subGuassian random variables:

**Claim C.1** (Maximum of a sequence of subGuassian random variables.). *For a sequence of $\sigma$-subGaussian random variables $X_1, \dots, X_n \in \mathbb{R}$. We have that with probability at least $1 - \delta$,*

$$\max_i |X_i| \le \sqrt{2}\sigma(\sqrt{\log n} + \sqrt{\log(1/\delta)}))$$

*Specifically, under the condition that $\delta \gtrsim 1/n$, we have*

$$\max_i |X_i| \lesssim \sigma\sqrt{\log n}.$$

*Proof of Claim C.1.* By taking union bound, we have

$$
\mathbb{P}(\max_i X_i \geq u) = \mathbb{P}(\exists i : X_i \geq u)
$$
$$
\leq n\mathbb{P}(X_1 \geq u)
$$
$$
\leq n\exp(-\frac{u^2}{2\sigma^2}).
$$

By setting $u = \sqrt{2}\sigma(\sqrt{\log n} + \sqrt{\log(1/\delta)})$, we have

$$
\mathbb{P}(\max_i X_i \geq \sqrt{2}\sigma(\sqrt{\log n} + \sqrt{\log(1/\delta)}))
$$
$$
\leq n\exp(-\frac{\sqrt{2}\sigma(\sqrt{\log n} + \sqrt{\log(1/\delta)})^2}{2\sigma^2})
$$
$$
= n\exp(-(\sqrt{\log n} + \sqrt{\log(1/\delta)})^2)
$$
$$
= \exp(-\log(1/\delta) - \sqrt{\log(n)\sqrt{1/\delta}})
$$
$$
\leq \delta.
$$

Therefore, with probability at least $1 - \delta$, we have

$$
\max_i |X_i| \leq \sqrt{2}\sigma(\sqrt{\log n} + \sqrt{\log(1/\delta)})
$$

$\square$

The next claim bounds the norm of sum of epsilon.

**Claim C.2.** *Under Assumption [A4] which assumes that $\epsilon_i$ are subGaussian, we have that with probability at least $1 - \delta$,*

$$
\left\| \frac{1}{n} \sum_{i\in[n]} \epsilon_i \right\| \leq 4\sigma\sqrt{\frac{p}{n}} + 2\sigma\sqrt{\frac{\log(1/\delta)}{n}}.
$$

*Specifically, under the condition that $\delta \gtrsim \exp(-4p)$, it holds that*

$$
\left\| \frac{1}{n} \sum_{i\in[n]} \epsilon_i \right\| \lesssim \sigma\sqrt{\frac{p}{n}}.
$$

*Proof.* Note that since $\epsilon_i$ is $\sigma^2$-subGaussian, its summation $\sum_{i\in[n]} \epsilon_i$ is $n\sigma^2$-subGaussian.

Therefore, we have that

$$
\frac{1}{n}\left\| \sum_{i\in[n]} \epsilon_i \right\| \leq 4\sigma\sqrt{\frac{p}{n}} + 2\sigma\sqrt{\frac{\log(1/\delta)}{n}}
$$

$\square$

The next claim shows that the number of noisy data is approximately in the same order with $\rho n$.

**Claim C.3** (Number of noisy data). *Denote $|\mathcal{K}|$ as the number of samples with flipped labels. With probability at least $1 - \delta$, there exists a constant $c$ such that*

$$
||\mathcal{K}| - \rho n| \leq \sqrt{cn\log(2/\delta)}.
$$

*Furthermore, when $\delta \gtrsim 2\exp(-\frac{1}{8}\rho^2 n)$, we have that with probability at least $1 - \delta$,*

$$
|\mathcal{K}| \gtrsim \frac{1}{2}\rho n \text{ and } |\mathcal{K}| \lesssim \frac{3}{2}\rho n.
$$

*Proof.* Note that $\mathcal{K} = \sum_i u_i$, where $u_i$ are drawn from independent Bernoulli distribution with parameter $\rho$. Therefore, by Hoeffding's inequality, there exists constant $c$ such that with probability $1 - 2\exp\left(-\frac{ct^2}{n}\right)$,

$$||\mathcal{K}| - \rho n| \leq t.$$

By setting $t = \sqrt{\frac{n}{c}\log(\frac{2}{\delta})}$, we have that with probability at least $1 - \delta$,

$$||\mathcal{K}| - \rho n| \leq \sqrt{\frac{n}{c}\log(\frac{2}{\delta})}.$$

We then rewrite the constant $c$ to reach the above conclusion. $\square$

The next Claim C.4 shows that analyzing $\mathcal{L}_{01}$ loss on noiseless data is sufficient under noisy regimes.

**Claim C.4.** *Under the noisy regime, the 0-1 population loss on noiseless data $\mathcal{L}_{01}(\boldsymbol{w}) = \mathbb{P}(y\boldsymbol{x}^\top \boldsymbol{w} < 0)$ has the same order with the 0-1 population loss (excess risk) on noisy data $\tilde{\mathcal{L}}_{01}(\boldsymbol{w}) - \eta = \mathbb{P}(\tilde{y}\boldsymbol{x}^\top \boldsymbol{w} < 0) - \eta$. Therefore, we analyze $\mathcal{L}_{01}(\boldsymbol{w})$ in this paper as a surrogate of $\tilde{\mathcal{L}}_{01}(\boldsymbol{w})$.*

*Proof.* Note that we here consider the $\mathcal{L}_{01}$ on the noiseless data, and one can get $\tilde{\mathcal{L}}_{01}$ on the noisy data by just adding $\rho$ to $\mathcal{L}_{01}$. That is to say,

$$\begin{aligned}
\tilde{\mathcal{L}}_{01} =& \mathbb{P}(\tilde{y}\boldsymbol{x}^\top \boldsymbol{w} < 0) \\
=& \eta\mathbb{P}(-y\boldsymbol{x}^\top \boldsymbol{w} < 0) + (1 - \eta)\mathbb{P}(y\boldsymbol{x}^\top \boldsymbol{w} < 0) \\
=& \eta(1 - \mathcal{L}_{01}) + (1 - \eta)\mathcal{L}_{01} \\
=& \eta + (1 - 2\eta)\mathcal{L}_{01},
\end{aligned}$$

And therefore $\tilde{\mathcal{L}}_{01} - \eta$ has the same order with $\mathcal{L}_{01}$. $\square$

# D EXPERIMENTS

## D.1 IMPLEMENTED TASKS

We conduct experiments on three public available datasets along with synthetic GMM data. First, we briefly introduce each dataset.

**ImageNet** is an image classification dataset containing 1.2 million pictures belonging to 1000 different classes. **CIFAR10** is an image classification dataset containing 60000 pictures belonging to 10 different classes. **Penn Tree Bank(PTB)** is a dataset containing natural sentences with 10000 different tokens. We use this dataset for language modelling task. **Synthetic GMM data** is generated following the setup in Section 3.1. We generate data for $n \in \{2^4, ..., 2^9\}, p \in \{2^4, ..., 2^9\}, \rho \in \{0, 0.4\}, |\mu| = 40, \sigma = 1, n \leq p$.

We then describe experiment details for each dataset.

For **ImageNet**, we train a ResNet50 from scratch. We use standard cross entropy loss. We first train the model for 90 epochs using $SGD$ as optimizer, with initial learning rate $1e-1$, momentum $0.9$ and weight decay $1e-4$. The learning rate will decay by a factor of 10 for every 30 iterations. Then we train the model for another 410 epochs, with initial learning rate $1e-3$, momentum $0.9$ and weight decay $1e-4$ and learning rate will decay by a factor of 1.25 for every 50 iterations.

For **CIFAR10**, we randomly flip the label with probability $\{0, 0.1, 0.2, 0.3, 0.4, 0.5, 0.6\}$ and for each train a ResNet18 from scratch. We use standard cross entropy loss. We train each model for 200 epochs using $SGD$ as optimizer, with learning rate $1e-1$, momentum $0.9$ and weight decay $5e-4$. We use a cosine learning rate decay scheduler. For ablation study, we fix the flip probability to be $0.2$ and train WideResNet16-$W$ on the noisy dataset with $W \in 1, 1.25, 1.5, 1.75, 2, 3, 4$. We repeat each experiments for four times. We train each model for 200 epochs using $SGD$ as optimizer, with learning rate $1e-1$, momentum $0.9$ and weight decay $5e-4$. We use a cosine learning rate decay scheduler

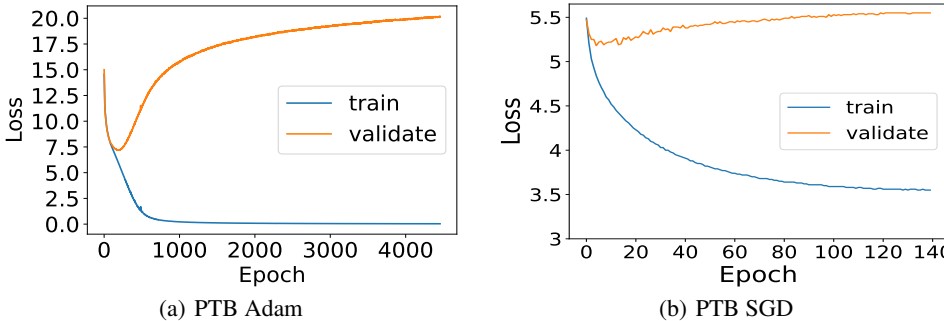

(a) PTB Adam

(b) PTB SGD

Figure 5: **Overfitting Behavior for language modelling.** We use transformer to train language modelling on Penn Tree Bank and plot the training loss as well as validation loss. We find that transformer overfits Penn Tree Bank significantly.

For **Penn Tree Bank**, we train a standard transformer from scratch. We train one model for 4800 epochs using $ADAM$ as optimizer, with learning rate 5e$-$4, beta $(0.9, 0.98)$ and weight decay 1e$-$2. We use an inverse square learning rate scheduler.We train another model for 140 epochs using $SGD$ as optimizer with learning rate 5.0. We use a step learning rate schedule with step size 1 and gamma $0.95$.

For **Synthetic GMM data**, we train a linear classifier for each dataset, initialized from $0$. We use logistic loss. We train each model using $SGD$ as optimizer with learning rate 1e$-$5 until training loss decrease below $0.05$.

## D.2 OVERFITTING ON PTB WITH LANGUAGE MODELLING

In this subsection, we show that the training a small Transformer on PTB for language modelling also leads to overfitting, as shown in Figure 5. This is expected based on our analysis. If we view language modelling as a classification task: predicting the next word based on a prefixed sequence, then the ground truth label will naturally be noisy.

## D.3 MORE GMM EXPERIMENTS

We hereby present more GMM experiments results. We consider signal-to-noise ratio between $\{10, 20, 40, 80, 160\}$ and the result is shown in figure 6. Notice that according to our theory, we can roughly separate the GMM models into three cases ignoring some technical conditions.

Case 1 $\|\mu\|/\sigma = o((p/n)^{1/4})$ which corresponds to the case where [A6] doesn't hold. In this case, as the signal-to-noise ratio is too small, even on noiseless data, the model will suffer a constant excess risk.

Case 2 $\|\mu\|/\sigma = O((p/n)^{1/4} + \sqrt{\log n})$ while $p = \Theta(n)$. In this case the signal-to-noise ratio is great enough, however the overparameterization is mild, hence although the model can benign overfit on noiseless data, it will fail to benign overfit noisy data. However early stopping on the noisy data will still gives a classifier that have nonvacuous generalization guarantee.

Case 3 $\|\mu\|/\sigma = O((p/n)^{1/4} + \sqrt{\log n})$ while $p = \omega(n)$. In this heavy overparameteized regime, the model will benignly overfit both the noisy and clean data.

Our simulation aligns with the theoretical prediction as

(1) For small signal-to-noise ratio as 5, we observe that when $p/n$ grows to $2^7$, the linear model starts to suffer an constant excess risk even for the noiseless data. For the noisy data, we need a larger signal-to-noise ratio compared with the overparameterization. This corresponds to Case 1 in our prediction and explain the constant excess risk in Figure 6(a) and the

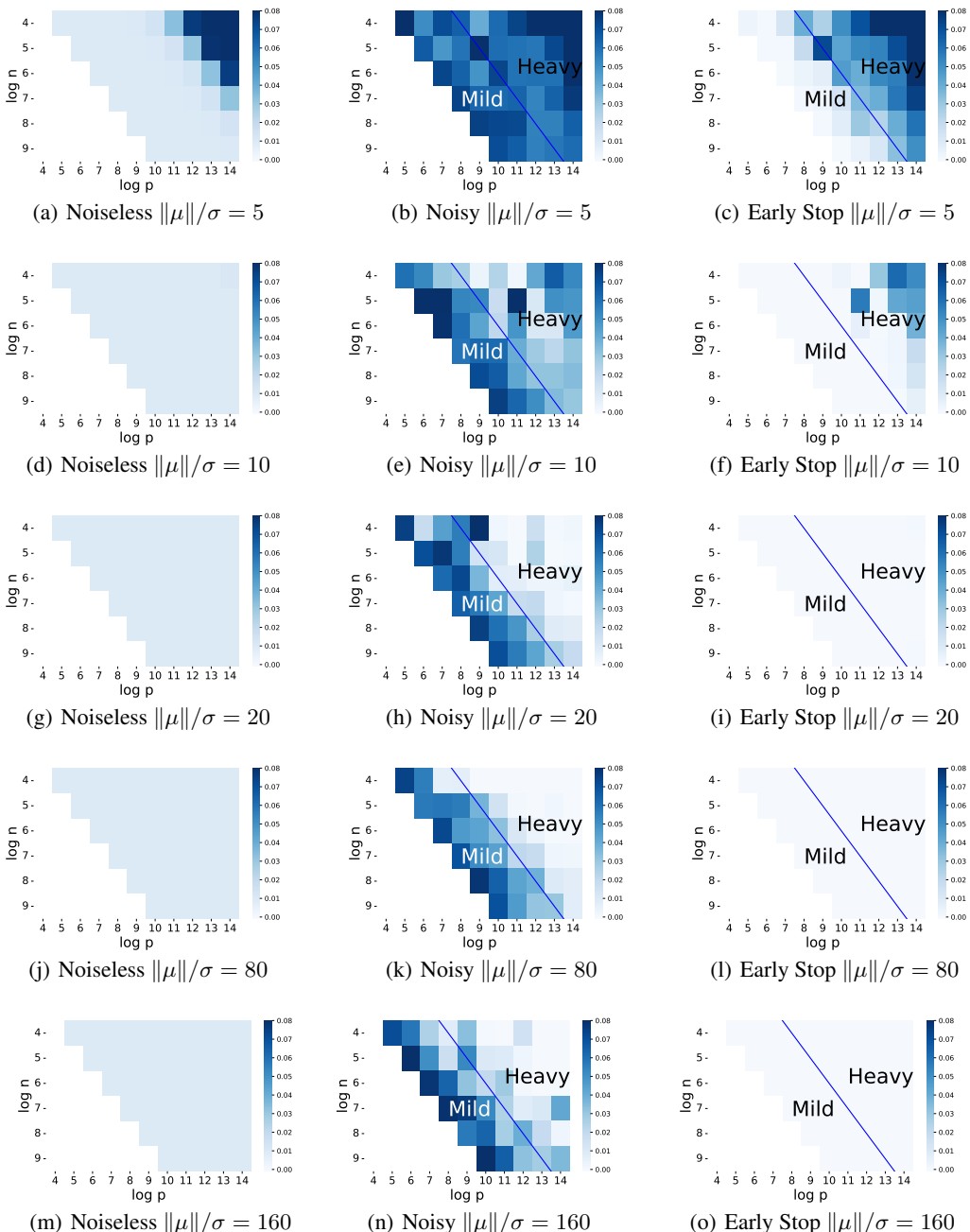

Figure 6: **Extended GMM Experiments.** The figure is derived in the same way as Figure 2.

overfitting in the heavy overparameterized regime in Figure 6(b), Figure 6(c), Figure 6(e) and Figure 6(f).

(2) We observe similar effect as in Figure 2 for $\frac{\|\mu\|}{\sigma}$ spanning across $\{10, 20, 40, 80, 160\}$ that in the mild overparameterized regime, non-benign overfitting happens consistently, while for the heavy overparameterized regime, benign overfitting takes place.

(3) We also notes that using technique as early stopping prevents overfitting and gives a classifier with generalization guarantee for $\frac{\|\mu\|}{\sigma}$ spanning across $\{10, 20, 40, 80, 160\}$ as predicted.

# E  DISCUSSION

## E.1  COMPARISON BETWEEN REGRESSION AND CLASSIFICATION

Although Bartlett et al. (2020) shows similar findings in regression regimes, we emphasize that the techniques can be pretty different. The difference between regression and classification problems can be split into two folds:

**(a) technical difference.** Regression problem usually permits closed-form solution through matrix pseudo-inverse while classification problem does not. Therefore, one needs to use other terms, *e.g.*, margin, to study the generalization bound. Besides, the commonly-used operations in regression settings, *e.g.*, variance-bias decomposition do not apply in classification settings. Therefore, we need some different techniques during the proof. For example, when proving Theorem 3.1.2 we need to choose those samples with wrong labels, which is impossible in regression regimes.

**(b) different label noise type.** In regression problems, the label noise is usually subGaussian type. However, the label noise is totally different in classification problems and therefore their results cannot be extended to the classification settings. However, as the most commonly-used learning task, the analysis of classification is important.

## E.2  DETECTING BENIGN OVERFITTING IN PRACTICE

Benign overfitting refers to that *large models can fit data well without giving poor generalization performance* (Bartlett et al., 2021), indicating the models operate outside the realm of uniform convergence. There could be multiple mathematical instantiations of the statement. For example, Bartlett et al. (2020) bound the generalization error of the last-epoch model with sample size and model size to characterize benign overfitting, and prove that the validation accuracy approaches Bayesian optimal as sample size increases. Our theoretical claims (Statements 1 and 3 of Theorem 3.1) follow the same criterion. We show that the validation error of a fully trained linear model approaches the label error, and hence is Bayesian optimal.

However, the criterion *generalization error converges to zero* is hard to verify in experiments, as it is hard to determine when the model size and sample complexity are *large enough* for the generalization bound to be sufficiently small, given that the data set size is upper bounded. Hence we adopt two mild assumptions to test whether benign overfitting happens empirically.

1. Training the model for long-enough epochs (at least three times that of the standard practice) can work as an honest proxy for training for infinite epochs.

2. The gap of verification accuracy between the best-epoch model and the last-epoch model can work as an efficient proxy for the gap between the Bayesian optimal accuracy and the validation accuracy of the last-epoch model. This assumption is supported by our theoretical predictions (Statements 2 and 3 of Theorem 3.1).

Hence we adopt the different notion of benign overfitting, as the reviewer mentioned, 'validation performance does not drop while the model fits more training data points'. This notion is

1. coherent to the intuitive definition *embed all of the noise of the labels into the parameter estimate —without harming prediction accuracy*;

2. easily experimentally verifiable;

3. closely related to theoretical predictions in our work and previous works, in the sense that based on theoretical prediction, in the mild overparameterization regime (our work), the drop will be significant regardless of model size, and in the heavy overparameterization regime (previous works), the drop will diminish when the model size and sample complexity increases.

Finally, we would like to stress that the similarity between the overparameterization through model size and training a model longer is not required for our analysis.

