# OpenReview forum: "Benign Overfitting in Classification: Provably Counter Label Noise with Larger Models"
_ICLR.cc/2023/Conference — ICLR 2023 poster_

### Official Review · Reviewer_P3FZ · 2022-10-19

**Confidence:** 4
**Correctness:** 3
**Technical Novelty And Significance:** 3
**Empirical Novelty And Significance:** 3
**Recommendation:** 6

**Clarity, Quality, Novelty And Reproducibility:**

The paper is clearly organized and well-written.
The core idea is also novel, to consider the label noise classification setting with mild parameterization.
I checked the proof and it generally holds, except for some issues I mentioned above.

**Strength And Weaknesses:**

Strength:
1. Paper is well-written and easy to follow.
2. The core idea is demonstrated with both theory and experiments.

Weakness:
1. “fast convergence rate in Statement One and Statement Three”: forgive my ignorance, I only see the bound on the loss but did not see any results related to the convergence rate. For example, any bound on the difference between $L(t+1)$ and $L(t)$? Is it a linear convergence?
2. For separable data in the noisy setting, are you considering the flipped labels or their clean labels? I suppose one cannot have noisy data separable on both their flipped labels and their clean labels. If you consider the noisy data with flipped labels to be separable, is this conflict with the noiseless setting? I.e., you make separable noiseless data into noisy by flipping some labels, which will give you inseparable noisy data.
3. On page 16 case 2, “distance between $\mu$ and $\bar{\mathbf{x}}_\kappa^{\top}$ must be less than the distance from $\mu$ to its projection on the separating hyperplane”. Does that imply $|\mu^\top \mathbf{w}| \geq \| \bar{\mathbf{x}}_\kappa^{\top} − \mu \|$?
4. In proof of Thm 5.3, I suppose the bounds over $\max_{t \in [T]}$ (Eq. 10, 12, 15) may not be tight at the same $t$?
5. To complete the argument in this work, I suppose you should also show that with increasing parameterization, the overfitting becomes benign (e.g. the lower bound of risk becomes diminished)? Current benign overfitting papers do not have this conclusion in label noise classification setting.

**Summary Of The Paper:**

This work analyzes the benign overfitting when the logistic regression is mild over-parameterization, in the setting of label noise. The core theory results are: 1) excess risk can be lower bounded at infinite SGD iteration; 2) the risk with early stopping can be upper bounded by 1/n (the number of samples. The authors also empirically verify the classification error under label noise and with different network widths.

**Summary Of The Review:**

In general, I think this is a good paper with solid contributions to both theory and experiments.
I hope the authors could address my concerns above.

---

> ### Author Response · Authors · 2022-11-09
> **Response to reviews**
>
> We thank the reviewer for the detailed and constructive comments, which help us polish our manuscript. Below, our response to the reviewer's questions is below.
>
> >“fast convergence rate in Statement One and Statement Three”: forgive my ignorance, I only see the bound on the loss but did not see any results related to the convergence rate. For example, any bound on the difference between L(t+1) and L(t)? Is it a linear convergence?
>
> We apologize for the confusion. The convergence rate here is with respect to the sample size $n$. Besides, the fast convergence rate here means the rate $n^{-c}$. We have added a footnote here (Page 6) to make it clearer.
>
> >For separable data in the noisy setting, are you considering the flipped labels or their clean labels? I suppose one cannot have noisy data separable on both their flipped labels and their clean labels. If you consider the noisy data with flipped labels to be separable, is this conflict with the noiseless setting? I.e., you make separable noiseless data into noisy by flipping some labels, which will give you inseparable noisy data.
>
> We thank the reviewer for the insightful question. Both the noiseless and noisy labels can be separable under overparameterization (the separation hyperplane can be different). For example, if $XX^\top$ is full-rank (where $X \in \mathbb{R}^{n \times p}$ denotes the design matrix, $n$ and $p$ denote the sample size and dimension, respectively), one can always separate all the samples given any labels (in both noisy and noiseless regimes).
>
> >On page 16 case 2, “distance between $\mu$ and $\bar{x}_K^\top$ must be less than the distance from $\mu$ to its projection on the separating hyperplane”. Does that imply $|\mu^\top w| \geq |\bar{x}_K^\top -\mu|$?
>
> Yes, the argument indeed implies $|\mu^\top w| \geq |\bar{x}_K^\top -\mu|$.
>
> >In proof of Thm 5.3, I suppose the bounds over $\max_{t \in [T]}$ (Eq. 10, 12, 15) may not be tight at the same t?
>
> Yes, the optimal time $t$ for these terms can be different.
>
> >To complete the argument in this work, I suppose you should also show that with increasing parameterization, the overfitting becomes benign (e.g. the lower bound of risk becomes diminished)? Current benign overfitting papers do not have this conclusion in label noise classification setting.
>
> We thank the reviewer for the insightful and nice comments!
>
> One way to achieve overparameterization is to send $r=p/n$ to infinity. Indeed, the lower bound of risk becomes diminished with increasing parameterization. However, completing the argument requires an upper bound, and the proof becomes much more complex under this mild-overparameterization regime. The reason is that: we need a surrogate classifier with a large margin during the proof, yet the argument breaks done in this mild-overparameterization regime. We leave the upper bound for future work.
>
> Alternatively, if $p = \Omega (n \log n)$, you are correct that with increasing overparameterization, the overfitting will become benign. The results are indeed shown in previous works [1] and [2].
>
> [1] Niladri S. Chatterji and Philip M. Long. Finite-sample analysis of interpolating linear classifiers in the overparameterized regime.
>
> [2] Ke Wang and Christos Thrampoulidis. Benign overfitting in binary classification of gaussian mixtures.
>
> Once again, we appreciate the reviewer's precious time. We are eager to engage in further discussions to clear out any confusion.

---

### Official Review · Reviewer_k2R8 · 2022-10-25

**Confidence:** 3
**Correctness:** 3
**Technical Novelty And Significance:** 3
**Empirical Novelty And Significance:** 3
**Recommendation:** 6

**Clarity, Quality, Novelty And Reproducibility:**

* The paper is very clear. Very organized and enjoyable to read.

* The paper is of high quality.

* The paper is quite novel to me.

* Question to authors:
   + It seems that the definition of benign overfitting in this work is different from [2], where they define benigh overfitting as  deep neural networks seem to predict well, even with a perfect fit to noisy training data. Can you provide a discussion on this?

* Minors
   + Page 3 missing space between assumptions and (instead of

[2] Bartlett, Peter L., et al. "Benign overfitting in linear regression." Proceedings of the National Academy of Sciences 117.48 (2020): 30063-30070

**Strength And Weaknesses:**

**Strength**

* This work studies benign overfitting in the mild over-parameterziation regime which is novel and interesting compared to previous works.

* The authors further study the noisy case and present the correspoding results. They find that mild-overparameterization is not enough for classification noisy problem. This result seems important to me.


**Weaknesses**

* This work directly use the result from implicit bias theory (Proposition 3.1). It is unclear what will happen when we study the gradient descent training like [1].

[1] Cao, Y., Chen, Z., Belkin, M., & Gu, Q. (2022). Benign Overfitting in Two-layer Convolutional Neural Networks. arXiv preprint arXiv:2202.06526.

**Summary Of The Paper:**

This work studies benign overfitting with respect to the level of over-parameterization and whether under a noisy regime through implicit bias theory. This study is inspired by the phenomenon that the ResNet model overfits benignly on Cifar10 but not benignly on ImageNet. The authors identify a phase change: unlike in the previous heavy overparameterization settings, benign overfitting can now fail in the presence of label noise. The experimental simulation verifies the theoretical results.


**Summary Of The Review:**

The paper presented a relatively novel idea (to me) on benign overfitting. It presented a set of thorough theoretical analyses and experiments to validate the method. I tend to accept this paper.

---

> ### Author Response · Authors · 2022-11-09
> **Response to reviews**
>
> We thank the reviewer for the insightful and constructive comments, which help us polish our manuscript. Below, we do our best to address the reviewer's questions adequately.
>
> >This work directly use the result from implicit bias theory (Proposition 3.1). It is unclear what will happen when we study the gradient descent training like Cao et al.(2022).
>
> We thank the reviewer for the comment. We could indeed obtain similar results for all our statements under gradient descent since gradient descent has a similar max-margin implicit bias to Proposition 3.1 [1].
> However, the possible extension to neural networks as in Cao et al.(2022) [2] is still unclear and we will consider that in future works.
> We have added this discussion in the new version.
>
> [1] Soudry, Daniel, Elad Hoffer, Mor Shpigel Nacson, Suriya Gunasekar, and Nathan Srebro. "The implicit bias of gradient descent on separable data." The Journal of Machine Learning Research 19, no. 1 (2018): 2822-2878.
>
> [2] Cao, Y., Chen, Z., Belkin, M., & Gu, Q. (2022). Benign Overfitting in Two-layer Convolutional Neural Networks. arXiv preprint arXiv:2202.06526.
>
> >It seems that the definition of benign overfitting in this work is different from [2], where they define benigh overfitting as deep neural networks seem to predict well, even with a perfect fit to noisy training data. Can you provide a discussion on this?
>
> We apologize for not making everything clear. In words, benign overfitting refers to the fact that large models can fit data well without giving poor generalization performance [1], indicating that the models operate outside the realm of uniform convergence. There could be multiple mathematical instantiations of the statement. For example, Bartlett et al. [2] bound the generalization error of the last-epoch model with sample size and model size to characterize benign overfitting, and prove that the validation accuracy approaches Bayesian optimal as sample size increases. Our theoretical claims (Statements 1 and 3 of Theorem 3.1) follow the same criterion. We show that the validation error of a fully trained linear model approaches the label error, and hence is Bayesian optimal.
>
> ***However, the criterion 'generalization error converges to zero' is hard to verify in experiments***, as it is hard to determine when the model size and sample complexity are large enough for the generalization bound to be sufficiently small, given that the data set size is upper bounded. Hence we adopt two mild assumptions to test whether benign overfitting happens empirically.
> 1. Training the model for long-enough epochs (at least three times that of the standard practice) can work as an honest proxy for training for infinite epochs.
> 2. The gap of verification accuracy between the best-epoch model and the last-epoch model can work as an efficient proxy for the gap between the bayesian optimal accuracy and the validation accuracy of the last-epoch model. This assumption is supported by our theoretical predictions (Statements 2 and 3 of Theorem 3.1).
>
> Hence we adopt the different notion of benign overfitting, as the reviewer mentioned, 'validation performance does not drop while the model fits more training data points'. This notion is
> 1. coherent to the intuitive definition 'embed all of the noise of the labels into the parameter estimate —without harming prediction accuracy';
> 2. easily experimentally verifiable;
> 3. closely related to theoretical predictions in our work and previous works, in the sense that based on theoretical prediction, in the mild overparameterization regime (our work), the drop will be significant regardless of model size, and in the heavy overparameterization regime (previous works), the drop will diminish when the model size and sample complexity increases.
>
> Again, we thank the reviewer for this comment. We have included the above discussion in our manuscript.
>
> [1]Peter L Bartlett, Andrea Montanari, Alexander Rakhlin. Deep learning: a statistical viewpoint.
>
> [2]Peter L Bartlett, Philip M Long, Gábor Lugosi, and Alexander Tsigler. Benign overfitting in linear regression.
>
> Once again, we appreciate the reviewer's precious time. We are eager to engage in further discussions to clear out any confusion.

---

### Official Review · Reviewer_NJij · 2022-10-25

**Confidence:** 2
**Correctness:** 3
**Technical Novelty And Significance:** 3
**Empirical Novelty And Significance:** 2
**Recommendation:** 6

**Clarity, Quality, Novelty And Reproducibility:**

Clarity:
* I believe in general the clarity of this paper needs to be improved to help the reader digest the result, as noted above.
* Some notations are not defined in this paper. For example,  $\mu$ in Section 3.1. The definition of notations is overall sloppy in this section. It is better to have a dedicated section presenting the notations.
* Some terms are not defined in this paper. For example, in Section 3.3, quote "The strange phenomenon happens because we distinguish the randomness in the training set from the randomness in the test set.". But what exactly is "randomness" here? I believe more discussion on the "difference of the randomness" is required here.

Quality and Novelty:
* I think the contribution of this work depends on whether the specific setting studied in this paper is novel compared to previous works. I am not that familiar with the related works. But I am a little bit worried that the mild-overparameterization regime may already be presented in previous works as a side result. I would be happy if the authors can correct me.














**Strength And Weaknesses:**

Strength:
* This paper is well-organized and focused on an important problem in understanding the generalization of deep neural models. The claim is overall clear and the target setting is specific. Some observations made in this paper are also interesting such as deep models on ImageNet and CIFAR-10 have different overfitting phenomena.

Weakness:
* I believe the clarity of the paper needs to be improved. First, the definition of benign overfitting is rather ambiguous in this paper. I may not be an expert in this field, but based on my understanding, benign overfitting in existing works studies the generalization error as model size increases. However, in the introduction of this paper, benign overfitting is termed as 'validation performance does not drop while the model fits more training data points'. It appears to me that the authors try to study the dynamics of generalization error through model training (i.e. epoch-wisely). This is further corroborated by Figure 1 where the performance curves through training are shown. While I understand overparameterization through model size may bear similarities in training a model longer, it would be better if the authors can make it explicit.

* In the experiment section, the above problem becomes more prominent. For example, in Figure 3 the authors show the performance curves with respect to epochs, while in Figure 4 the authors show the performance curves with respect to model size.

* I might miss something in Theorem 3.1. But why does the lower bound to the generalization error in a noisy setting (Statement 2) show benign overfitting fails? I believe a lower bound to the generalization error should also exist in a noiseless setting.


**Summary Of The Paper:**

This paper studies the benign overfitting phenomenon of deep neural models, namely over-parameterized deep neural models fitting the training data can still achieve low generalization error. The authors focus on a specific regime where the model is mildly overparameterized and label noise is prevalent. They show that in such a regime, benign overfitting may fail. This explains the observation that ResNet model overfits benignly on CIFAR-10 but not on ImageNet, since ImageNet may be noisier. They also show that early stopping can help avoid overfitting in such a regime.

**Summary Of The Review:**

The promises and observations made in this paper are interesting but it would be hard to judge the contribution right now given the clarity issue. I would be willing to increase my score if the authors can address my above concerns.

---

> ### Author Response · Authors · 2022-11-09
> **Response to reviews**
>
> We thank the reviewer for the insightful and supportive comments. Below, we do our best to address the reviewer's questions adequately. To start, we highlight that in our setup, the number of training epochs is not related to the level of overparameterization. Instead, it's aimed at demonstrating that overfitting is not benign. We hope the following details can clarify this confusion.
>
> >I believe the clarity of the paper needs to be improved. First, the definition of benign overfitting is rather ambiguous in this paper. [...] benign overfitting in existing works studies the generalization error as model size increases. However, in the introduction of this paper, benign overfitting is termed as 'validation performance does not drop while the model fits more training data points'. It appears to me that the authors try to study the dynamics of generalization error through model training (i.e. epoch-wisely). This is further corroborated by Figure 1 where the performance curves through training are shown. While I understand overparameterization through model size may bear similarities in training a model longer, it would be better if the authors can make it explicit.
>
> We apologize for not making everything clear. In words, benign overfitting refers to the fact that *large models can fit data well without giving poor generalization performance* [1], indicating that the models operate outside the realm of uniform convergence. There could be multiple mathematical instantiations of the statement. For example, Bartlett et al. [2] bound the generalization error of the last-epoch model with sample size and model size to characterize benign overfitting, and prove that the validation accuracy approaches Bayesian optimal as sample size increases. Our theoretical claims (Statements 1 and 3 of Theorem 3.1) follow the same criterion. We show that the validation error of a fully trained linear model approaches the label error, and hence is Bayesian optimal.
>
> ***However, the criterion 'generalization error converges to zero' is hard to verify in experiments***, as it is hard to determine when the model size and sample complexity are *large enough* for the generalization bound to be sufficiently small, given that the data set size is upper bounded. Hence we adopt two mild assumptions to test whether benign overfitting happens empirically.
> 1. Training the model for long-enough epochs (at least three times that of the standard practice) can work as an honest proxy for training for infinite epochs.
> 2. The gap of verification accuracy between the best-epoch model and the last-epoch model can work as an efficient proxy for the gap between the bayesian optimal accuracy and the validation accuracy of the last-epoch model. This assumption is supported by our theoretical predictions (Statements 2 and 3 of Theorem 3.1).
>
> Hence we adopt the different notion of benign overfitting, as the reviewer mentioned, 'validation performance does not drop while the model fits more training data points'. This notion is
> 1. coherent to the intuitive definition 'embed all of the noise of the labels into the parameter estimate —without harming prediction accuracy';
> 2. easily experimentally verifiable;
> 3. closely related to theoretical predictions in our work and previous works, in the sense that based on theoretical prediction, in the mild overparameterization regime (our work), the drop will be significant regardless of model size, and in the heavy overparameterization regime (previous works), the drop will diminish when the model size and sample complexity increases.
>
> Again, we thank the reviewer for this comment. We have included the above discussion in our manuscript.
>
> [1] Peter L Bartlett, Andrea Montanari, Alexander Rakhlin. Deep learning: a statistical viewpoint.
>
> [2] Peter L Bartlett, Philip M Long, Gábor Lugosi, and Alexander Tsigler. Benign overfitting in linear regression.
>
>
> >In the experiment section, the above problem becomes more prominent. For example, in Figure 3 the authors show the performance curves with respect to epochs, while in Figure 4 the authors show the performance curves with respect to model size.
>
> We apologize for the confusion. Figure 3 and Figure 4 show the importance of label noise and overparameterization level, respectively. These are the two factors that influence benign overfitting, as our work emphasizes.
> In Figure 3, we show that adding noise would hurt generalization in the last-iterate classifiers, as our theory predicted. The plot against the epoch demonstrates that the performance at the last iterate dropped significantly compared to the best model (as a substitute for the Bayesian optimal model).
> In Figure 4, we want to emphasize that as the ***overparameterization ratio*** (width factor) increases, the last-iterate classifier generalizes better while the best-iterate classifier performs similarly, which accords with our theory.
> Therefore Figure 3 and Figure 4 have different illustrations.

---

> > ### Author Response · Authors · 2022-11-09
> > **Response to reviews (2)**
> >
> > >I might miss something in Theorem 3.1. But why does the lower bound to the generalization error in a noisy setting (Statement 2) show benign overfitting fails? I believe a lower bound to the generalization error should also exist in a noiseless setting.
> >
> > Benign overfitting means the generalization error goes to zero as the sample size (n) goes to infinity. A constant lower bound would break the requirements in benign overfitting, and therefore we claim that benign overfitting fails in noisy settings.
> > For the noiseless setting, although there must be a lower bound, such a lower bound will go to zero as the sample size goes to infinity (guaranteed by the upper bound).
> > This is the key difference between noisy and noiseless regimes.
> >
> > >I believe in general the clarity of this paper needs to be improved to help the reader digest the result [...] For example, in Section 3.3, quote "The strange phenomenon happens because we distinguish the randomness in the training set from the randomness in the test set.". But what exactly is "randomness" here?
> >
> > Sorry for the confusion. Here we mean that: the high probability guarantee is in order $1/n$ with respect to the randomness in the sampling of training set and the algorithm, and the 0-1 loss bound is in order $1/n^c$ where the 0-1 loss is equivalent to taking probability on test set. Although the bound $1/n^c$ can be much smaller than $1/n$ when $c$ is large, the 0-1 loss is in order $1/n$ after take union bound. We have revised the current manuscript to clarify this point. Thanks again for the advice.
> >
> > >I think the contribution of this work depends on whether the specific setting studied in this paper is novel compared to previous works.[...] But I am a little bit worried that the mild-overparameterization regime may already be presented in previous works as a side result. I would be happy if the authors can correct me.
> >
> > We thank the reviewer for the nice comments. As far as we know, this is the first paper to study the interaction between mild-overparameterization and label noise. Few existing works consider both two factors, as collected in Table 1. They differ from our work in two fundamental aspects.
> > Most previous works provide a generalization *upper* bound on the last iteration model. For example, previous works [1] and [2] provide results that the last iteration model benignly overfits the data in the heavy overparameterized regime. Our work provides a constant *lower* bound on the validation performance of the last iteration model, which is fundamentally different in both the result itself and the proof technique.
> > The results in [3] imply that the regression interpolator fails under mild overparameterization regimes while may work under heavy overparameterization. However, their results (a) did not sketch the role of label noise, and (b) mainly focused on regression settings where the interpolator has a closed-form solution. In the classification regime, such a closed form does not exist. Hence their proof techniques cannot be generalized to our setting.
> >
> > [1] Yuan Cao, Quanquan Gu, and Mikhail Belkin. Risk bounds for over-parameterized maximum margin classification on sub-gaussian mixtures.
> >
> > [2] Niladri S. Chatterji and Philip M. Long. Finite-sample analysis of interpolating linear classifiers in the overparameterized regime.
> >
> > [3] Peter L Bartlett, Philip M Long, Gábor Lugosi, and Alexander Tsigler. Benign overfitting in linear regression.
> >
> > Once again, we appreciate the reviewer's precious time. We are eager to engage in further discussions to clear out any confusion.

---

### Decision · Program_Chairs · 2023-01-20

**Decision:**

Accept: poster

**Justification For Why Not Higher Score:**

The reviewers are not very enthusiastic about this paper.

**Justification For Why Not Lower Score:**

N/A

**Metareview: Summary, Strengths And Weaknesses:**

This paper analyzes benign overfitting when the number of parameters is not significantly larger than the number of data points. Under this mild overparameterization setup, the analysis identifies a phase that benign overfitting can fail in the presence of label noise.

Strengths:
Interesting experimental observations.

Weaknesses:
The technical clarity needs improvement.


**Note From Pc:**

if the above contains the word "oral" or "spotlight" please see: "oral" presentation means -> notable-top-5% and "spotlight" means -> notable-top-25%. As stated in our emails, we are disassociating presentation type from AC recommendations

**Summary Of Ac-Reviewer Meeting:**

N/A